



# The diurnal Energy Balance Model (dEBM): A convenient surface mass balance solution for ice sheets in Earth System modeling

Uta Krebs-Kanzow[1], Paul Gierz[1], Christian B. Rodehacke[1,2], Shan Xu[1], Hu Yang[1], and Gerrit Lohmann[1]

[1]Alfred Wegener Institute Helmholtz Centre for Polar and Marine Research, Bremerhaven, Germany
[2]Danish Meteorological Institute, Copenhagen Ø, Denmark

**Correspondence:** Uta Krebs-Kanzow (uta.krebs-kanzow@awi.de)

**Abstract.** The surface mass balance scheme dEBM (diurnal Energy Balance Model) provides a novel interface between the atmosphere and land ice for Earth System modeling, which is based on the energy balance of glaciated surfaces. In contrast to empirical schemes, dEBM accounts for changes in the Earth's orbit and atmospheric composition. The scheme only requires monthly atmospheric forcing (precipitation, temperature, shortwave and longwave radiation, and cloud cover). It is also

computationally inexpensive, which makes it particularly suitable to investigate the ice sheets' response to long-term climate change. After calibration and validation, we analyze the surface mass balance of the Greenland Ice Sheet (GrIS) based on climate simulations representing two warm climate states: a simulation of the Mid Holocene (approximately 6000 years before present) and a climate projection based on an extreme emission scenario which extends to the year 2100. The former period features an intensified summer insolation while the 21st century is characterized by reduced outgoing long wave radiation.

Specifically, we investigate whether the temperature-melt relationship, as used in empirical temperature-index methods, remains stable under changing insolation and atmospheric composition. Our results indicate that the temperature-melt relation is sensitive to changes in insolation on orbital time scales but remains mostly invariant under the projected warming climate of the 21st century.

## 1 Introduction

At the surface, land ice gains mass through snow accumulation and loses mass through meltwater runoff and sublimation. The total surface mass balance (SMB) of a healthy ice sheet (i.e. not in the process of disintegration), needs to be positive in the long term, in order to compensate mass loss at the base, the peripheral surface, and the interfaces to oceans or proglacial lakes. The SMB exerts an essential control on the volume and geometry of ice sheets. Responding directly to climate change, the SMB substantially influences the waxing and waning of large-scale ice sheets in the course of glacial-interglacial cycles

on time scales of tens of thousands to 100,000 years (e.g. Hays et al., 1976; Huybers, 2006). The last glacial period was terminated by a rapid deglaciation, which caused the global sea level to rise by more than 100 m within 10,000 years (e.g. Lambeck et al., 2014) and resulted in a complete disintegration of the North American and Fennoscandian Ice Sheet (e.g. Peltier et al., 2015). In the present interglacial period, the Holocene, the Greenland Ice Sheet is the only ice sheet remaining on the Northern Hemisphere. Today, superimposed on the natural glacial-interglacial cycle, the anthropogenic climate change





will likely initiate an unprecedented, anthropogenic deglaciation. The Greenland Ice sheet is presently shrinking, and surface processes are predicted to amplify Greenland ice loss in the future (Oppenheimer et al., in press).

Ice sheet models forced by different climate projections predict a reduction in the mass of the Greenland Ice Sheet by the end of this cemtury, which could, according to high emission scenarios, contribute $9 \pm 5\,\mathrm{cm}$ to sea level rise from 2015 to

2100 (Goelzer et al., 2020). This assessment is in general agreement with earlier studies based on fewer ice sheet models and different SMB forcing (Rueckamp et al., 2019; Fuerst et al., 2015). In the 2002–2017 period, the Greenland ice sheet and surrounding glaciers contributed a total of $1\,\mathrm{cm}$ to sea level rise as measured by the Gravity Recovery and Climate Experiment (GRACE, Tapley et al., 2004). Reduced SMB explains more than half of the mass loss of the Greenland Ice Sheet (GrIS) (Sasgen et al., 2012). The change in SMB, primarily due to intensified meltwater runoff, has been attributed to positive air

temperature anomalies, a more extended melt period (e.g. Tedesco and Fettweis, 2012) and a reduction in cloud cover (Hofer et al., 2017). The GRACE observational period is characterized by several summers of extreme melt in Greenland and year to year changes in GrIS mass loss are large in comparison to the general acceleration over the full GRACE period. Specifically, the 2003–2013 period of accelerating mass loss and the subsequent deceleration are mostly associated with atmospheric circulation change (Greenland blocking, e.g. Fettweis et al., 2013; Bevis et al., 2019). To understand and predict the response of continental

ice sheets to a changing climate, it is critical to reliably diagnose the SMB component. A reliable estimate of the SMB can be produced with either (a) empirical approaches or from (b) consideration of the surface energy balance in physics-based schemes.

Empirically, the SMB of the GrIS can be estimated from near-surface air temperatures, for instance, by the positive degree-day method (Reeh, 1991). This particularly simple approach linearly relates mean melt rates to positive degree-days, PDD

(PDD refers to the temporal integral of near-surface temperatures (T) exceeding the melting point, e.g. Calov and Greve, 2005). Since this scheme has a low computational cost and is easy to handle, it has been widely used for long climate simulations (Charbit et al., 2013; Gierz et al., 2015; Heinemann et al., 2014; Roche et al., 2014; Ziemen et al., 2014) and paleo-temperature reconstructions (Box, 2013; Wilton et al., 2017). The PDD method was calibrated based on SMB observations from the GrIS and has demonstrated a good skill to reproduce recent changes in Greenland surface mass balance (Fettweis et al., 2020).

However, field measurements from glaciers outside of Greenland reveal that optimal parameters for the PDD scheme sharply differ for different latitudes, altitudes, or climate zones (Hock, 2003). Therefore it remains questionable if such empirical methods can be applied to climate projections of the next centuries or for the large northern hemispheric ice sheets of ice ages.

In contrast to empirical approaches, physics-based (and thus more universal) surface mass balance schemes for ice sheets and glaciers consider the sum of all energy fluxes $Q$ into the surface layer to calculate surface melt and refreezing of meltwater.

If the surface temperature is at the melting point, the melt rate is linearly related to the surface layer's net energy uptake. Refreezing is analogously related to a net heat release, but refreezing is limited by the amount of available liquid water. This asymmetry between melting and refreezing implicates that unresolved (spatial or temporal) variations of $Q$ around zero result in underestimation of meltwater runoff. Consequently, SMB calculations based on the energy balance should resolve the region where $Q > 0$ in summer, and should also resolve the diurnal melt-freeze cycle, which is particularly pronounced for clear

sky conditions. Away from their mostly steep margins, ice sheets usually rise to high elevations and are exposed to cold air



temperatures. Therefore, melting occurs in a narrow strip along the ice sheets' margins which requires a resolution of about 10km. This resolution is still beyond the scope of multidecadal global climate simulations or reanalysis products such as ERA-Interim (Dee et al., 2011). SMB estimates thus commonly involve some downscaling of coarse resolution forcing, either (i) dynamically through high-resolution regional climate models (e.g., MAR Fettweis et al. (2017), RACMO, Noel et al. (2018),

HIRHAM, Langen et al. (2015)), (ii) through the implementation of a one-dimensional SMB module in the climate model which recalculates the energy balance on different elevation classes (Vizcaino et al., 2010) or (iii) through downscaling of coarse resolution climate forcing according to the high-resolution topography for stand-alone SMB modeling (e.g. Born et al., 2019; Krapp et al., 2017). Overall, regional climate models perform best in comparison to observations as was demonstrated in the Greenland Surface Mass Balance Intercomparison Project (GrSMBMIP, Fettweis et al., 2020), which is primarily related

to a better representation of topographic precipitation. The computational cost of regional climate models prohibits from using these models on millennial timescales, which is necessary to study the slow response of ice sheets in a changing climate. To downscale SMB via elevation classes within Earth System Models is a relatively complex yet less costly approach and first applications yield promising results (e.g. Vizcaino et al., 2010; van Kampenhout et al., 2019). Its tight integration into an Earth System model prohibits its use as a flexible stand-alone SMB model. Stand-alone SMB models for long-term Earth

System modeling usually realize spatial downscaling by a lapse rate correction of coarse resolution temperatures to high-resolution topography. These efficient SMB schemes either involve empirical parameterizations which are not necessarily climate independent (Plach et al., 2018; de Boer et al., 2013) or usually require at least daily forcing. The BErgen Snow SImulator (BESSI, Born et al., 2019) uses a daily time step and considers the surface energy balance in combination with a sophisticated multi-layer snowpack model. BESSI appears to underestimate refreezing possibly because diurnal freeze-melt

cycles are not resolved. The Surface Energy and Mass balance model of Intermediate Complexity (SEMIC, Krapp et al., 2017) also uses a daily time step but statistically accounts for diurnal variations in surface temperature. Following a similar approach Krebs-Kanzow et al. (2018b) demonstrated that the diurnal melt period can be downscaled from monthly mean forcing by using the knowledge of the diurnal cycle of insolation at the top of the atmosphere, which is a function of latitude and season.

Here we refine the approach of Krebs-Kanzow et al. (2018b) and present a novel stand-alone SMB model, dEBM. The

presented model is efficient on millennial timescales and particularly suitable for Earth System modeling on long time scales in a modular framework such as Gierz et al. (2020), as it only requires monthly forcing. The scheme now also includes an albedo scheme, accounts for changes in atmospheric composition, and statistically resolves sub-monthly variability in cloud cover. In the first section of this paper, we provide a detailed model description. We then discuss the calibration of model parameters and evaluate the model against observations and a regional climate model. Finally, we apply dEBM with climate forcing from

a simulation of the Mid Holocene warm period, and from a transient climate simulation from the preindustrial period to the year 2100 based on the RCP8.5 scenario (Taylor et al., 2012). We specifically analyse the sensitivity of meltwater runoff to temperature change for these two distinct warm periods, to assess the validity of the empirical PDD method for different background climates. In the Appendix, tables A1 and A2 provide a list of parameters and variables used in the following.





## 2 Model Description

### 2.1 General concept

The dEBM is based on the surface energy balance (detailed in section 2.3) and simulates surface mass balance ($SMB$), melting ($ME$), refreezing ($RZ$), snowfall ($SF$), snow height ($SNH$), net runoff ($RO$), and albedo ($A$) at monthly time steps. As forcing,

the model requires monthly means of total precipitation ($PP_{cr}$), near-surface air temperature ($T_{cr}$), incoming surface short wave radiation ($SW_{cr}^{\downarrow}$), top of atmosphere (TOA) incoming shortwave radiation $\widehat{SW}$, incoming longwave radiation ($LW_{cr}^{\downarrow}$), and cloud cover ($CC_{cr}$). The suffix $_{cr}$ is given to the quantities, as usually, a coarse resolution climate model provides these forcing fields.

An essential part of the model is a spatial and temporal downscaling procedure of the forcing to increase the horizontal

resolution and to represent sub-monthly time scales (see section 2.4). In the following, $T$, $PP$, $SW^{\downarrow}$, $LW^{\downarrow}$ and $CC$ denote the respective monthly mean variables after downscaling or interpolating to the target grid. The spatial downscaling scheme involves a simple elevation correction of $T_{cr}$ and longwave radiation by applying a spatially and temporally constant lapse rate. The temporal downscaling parameterizes the diurnal freeze-melt cycle following Krebs-Kanzow et al. (2018b) and includes a statistic parameterization of the sub-monthly variability in cloud cover. Over the Greenland Ice Sheet daily cloudiness is

not normally distributed. Instead, it forms two distinct clusters, as apparent from an analysis of PROMICE automatic weather station data (Ahlstrom et al., 2008) on the GrIS (Fig. 1). Therefore, following a binary approach, we define two distinctively different radiative modes and name them "fair" and "cloudy".

After preparatory processing and downscaling of the forcing (as detailed in 2.4), we relate melting and refreezing to periods of positive and negative surface energy balance respectively. We separately evaluate the surface energy balance (section 2.3)

for cloudy and fair days, $Q_{cloudy}$ and $Q_{fair}$. This separation is introduced to account for the pronounced diurnal freeze-melt cycle of fair days. For this purpose we additionally estimate the energy balance of the daily melt and refreezing periods, $Q_{MP}$ and $Q_{fair} - Q_{MP}$ of fair days.

Here $_{MP}$ denotes quantities relevant during the melt period of fair days. The energy balance of the downscaled submonthly periods $Q_{cloudy}, Q_{MP}$ and $Q_{fair} - Q_{MP}$ yield respective melt or refreezing rates which contribute to the monthly surface mass

balance (section 2.2).

Another essential part of the model is the albedo scheme (section 2.6) which reflects a vibrant positive feedback: melting lowers the albedo of a snow surface, which in turn increases the energy uptake from shortwave radiation and re-intensifies melting. In consequence the ablation zone is distinguished by the lower albedo of wet snow and bare ice from higher albedos of dry and fresh snow in the accumulation zone. The dEBM distinguishes three surface types with distinct albedos: bright new

snow, dry snow, and dark wet show. The surface type of each grid point is assigned after an evaluation of the potential surface mass balance for each surface type, which implicates that the surface energy balance is preliminary calculated three times using the respective albedo values.

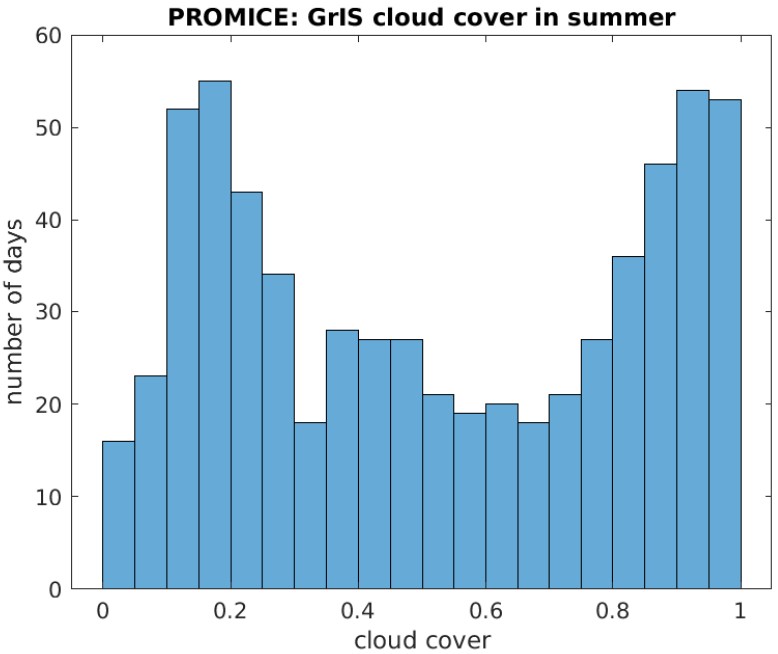

**Figure 1.** Histogram of daily cloud cover over the Greenland ice sheet throughout the summer months (June, July, August) based on daily measurements from up to 11 years of daily observations from 17 PROMICE weather station.

## 2.2 The surface mass balance

The main components determining the surface mass balance and the ice sheet's meltwater runoff $RO$

$$
\begin{aligned}
SMB &= SF - ME + RZ \\
RO &= ME + RF - RZ
\end{aligned}
\tag{1}
$$

are discussed individually in the following.

**snowfall** $SF$: $SF(PP, T)$ is a function of precipitation and near-surface air temperature as described in section 2.4.

**rainfall** $RF$: $RF(PP, T) = PP - SF$ is a function of precipitation and near-surface air temperature as described in section 2.4.

**Surface melt rate** $ME$: Melting is assumed to be only possible if near-surface temperature exceeds a minimum tempertature, $T_{min}$. As in Krebs-Kanzow et al. (2018b), we choose $T_{min} = -6.5\,°C$. Under melting conditions, melt rates of cloudy

days are linearly related to any positive net surface energy flux $max(0, Q_{cloudy})$ and the melt rate of fair days is related to $\max(0, Q_{fair}, Q_{MP})$, with $Q_{cloudy}, Q_{fair}$ being the surface energy balance of cloudy and fair days and $Q_{MP}$ being the energy balance during the sub-daily melt period of fair days (section 2.3). In most cases will $Q_{fair} < Q_{MP}$, as fair





days usually include a pronounced refreezing period at night (section 2.3). The total melt rate is

$$ME = \frac{1}{\rho L_f}((1 - CC)(\max(0, Q_{fair}, Q_{MP})) + CC \max(0, Q_{cloudy}))$$ (2)

with latent heat of fusion $L_f$ and the density of liquid water $\rho$.

**Refreezing rate** $RZ$ : Analogue to melting, we assume that $RZ$ is linearly related to negative net surface energy fluxes. The
5       maximum potential refreezing rate is

$$RZ_{pot} = \frac{1}{\rho L_f}(1 - CC)(\min(0, Q_{fair}, Q_{fair} - Q_{MP})) + CC \min(0, Q_{cloudy})$$ (3)

The total refreezing rate is bounded above by the amount of liquid water (from rainfall $RF$, see section 2.4 or melting
$ME$) and the storage capacity. Following the parameterization of Reeh (1991), we assume that the surface snow layer
can hold $60\,\%$ of its mass and the refreezing rate is

10       $$RZ = \min([(RF + ME), \frac{0.6 SNH(mth - 1)\rho_{water}}{\Delta t}, RZ_{pot}]).$$ (4)

$SNH$ is the water equivalent snow height, which is prognostic quantity; see section 2.7 for details, and $\Delta t$ is the prognos-
tic time step, which is here always a month. Melt water which is not refreezing within a month is added to the monthly
runoff.

Other contributions to the SMB such as sublimation, evaporation, and hoar are neglected by the dEBM, but could be estimated
independently if suitable forcing is available.

## 2.3 The surface energy balance

Following the approach in Krebs-Kanzow et al. (2018b), we consider the surface energy balance of a melting surface. The
energy balance of a melting surface can be simplified by applying the Stefan-Boltzmann law for longwave radiation with the
snow and ice surface temperature at melting point $T_i = T_0$. As surface temperature is not simulated by the dEBM, we define a
simple temperature criterion for the near-surface air temperature $T > T_{min}$ to identify potential melting conditions, we either
rule out melting from the outset or estimate melt rates from this simplified energy balance, depending on near-surface air
temperature ($T$) incoming shortwave radiation ($SW^{\downarrow}$), and albedo ($A(\text{SurfaceType})$) which is chosen according to the given
surface type (i.e., NewSnow, DrySnow, or WetSnow)

$$\begin{aligned} Q &= (1 - A(\text{SurfaceType}))SW^{\downarrow} + a(T - T_0) + b \\ a &= \epsilon_i \epsilon_a \sigma 4 T_0^3 + \beta \\ b &= -\epsilon_i \sigma T_0^4 + \epsilon_a \epsilon_i \sigma (T_0^4) + R \end{aligned}$$ (5)

where $\epsilon_i$ and $\epsilon_a$ are the longwave emissivities of ice and atmosphere, $\sigma$ is the Stefan-Boltzmann constant, the coefficient
$\beta$ represents the temperature sensitivity of the turbulent heat flux, and $T_0$ is the melting point. We use a constant $\epsilon_i = 0.98$
(Armstrong and Brun, 2008) and locally diagnose $\epsilon_a$ from the longwave radiation and air temperature forcing. We define that





all fluxes into the ice sheet's surface layer are positive. The term $R$ represents all unresolved energy fluxes, such as temperature-independent turbulent heat fluxes and heat conduction to the subsurface.

In contrast to Krebs-Kanzow et al. (2018b), parameters $a$ and $b$ are not constant because the atmospheric emissivity is diagnosed from longwave radiation and near-surface temperature. Since $Q_{fair}$ and $Q_{cloudy}$ are separately calculated (Equation

5), also the monthly shortwave radiation $SW^{\downarrow}_{cloudy/fair}$, albedo $A_{cloudy/fair}$ and atmospheric emissivity $\epsilon_{a,cloudy/fair}$ are differentiated between cloudy and fair condition (section 2.4).

Following Krebs-Kanzow et al. (2018b), we also consider the energy balance of the daily melt period of fair days, which is defined to be that part of a day when the elevation angle of the sun exceeds a critical value so that incoming shortwave radiation exceeds outgoing longwave radiation. In contrast to Krebs-Kanzow et al. (2018b), we estimate the critical elevation angle $\Phi$

for each location to account for the spatial variability in atmospheric emissivity $\epsilon_{a,fair}$. The energy balance of the daily melt period $Q_{MP}$ then is

$$Q_{MP} = \left( (1 - A)SW^{\downarrow}_{MP} + a_{MP}T_{MP} + b_{MP} \right) \tag{6}$$

where

$$
\begin{aligned}
T_{MP} &= PDD_{\sigma=3.5} \\
SW_{MP} &= \tfrac{\Delta t_\Phi}{\Delta t} \, q_\Phi SW_{fair} \\
a_{MP} &= \tfrac{\Delta t_\Phi}{\Delta t} \, \epsilon_i \epsilon_{a,fair} \sigma 4 T_0^3 + \beta \\
b_{MP} &= \tfrac{\Delta t_\Phi}{\Delta t} \left( -\epsilon_i \sigma T_0^4 + \epsilon_{a,fair} \epsilon_i \sigma(T_0^4) + R \right)
\end{aligned}
\tag{7}
$$

As in Krebs-Kanzow et al. (2018b), the near-surface temperature $T_{MP}$ during the melt period is represented by the always positive $PDD_{\sigma=3.5}$ (see 2.4). The ratio $\frac{\Delta t_\Phi}{\Delta t}$ is that fraction of a day when the sun exceeds the critical elevation angle, while $q_\Phi$ is the corresponding fraction of daily insolation which is effective during this period. The parameters $q_\Phi$ and $\Delta t_\Phi$ are functions of the elevation angle $\Phi$ (Krebs-Kanzow et al., 2018b), but here are estimated locally as we use spatially variable atmospheric emissivity.

## 20 2.4 Preprocessing of the climate forcing

The following downscaling steps are conducted prior to the actual SMB simulation to represent sub-monthly variability and not spatially resolved topographic features:

**Monthly mean atmospheric emissivity**

According to the Stefan-Boltzmann law longwave radiation can be expressed as a function of atmospheric emissivity and

temperature:

$$LW(\epsilon_a, T) = \epsilon_a \sigma T^4 \tag{8}$$

In preparation of the downscaling of longwave radiation we use Eq. 8 to diagnose $\epsilon_{a,cr}$ from coarse resolution downward longwave radiation and near-surface temperatures.





**Interpolation**

A bilinear interpolation between the source grid and the higher resolved target grid generates the fields of $H_{int}, T_{int}\, SW^{\downarrow}, \epsilon_a$, $PP, CC$ and $\widehat{SW}$.

**Spatial downscaling: lapse rate correction of air temperature**

We use a lapse rate of $\gamma = -0.007\,\mathrm{K\,m^{-1}}$ to transform the near-surface temperature to the surface elevation $H_{ice}$ of the target grid according to

$$T = T_{int} + \gamma(H_{ice} - H_{int}) \tag{9}$$

$H_{ice}$ may originate from an ice sheet simulation or reconstruction, and thus may differ substantially from the topography used in the climate model ($H_{int}$). The lapse rate corrected temperatures in combination with the interpolated $\epsilon_a$ can be used to
spatially downscale longwave radiation by applying the Stefan-Boltzmann law. This spatial downscaling of longwave radiation is here combined with a statistical downscaling of sub-monthly variability, as detailed below.

**Rain and snow**

Precipitation is partitioned into snowfall ($SF$) and rainfall ($RF$) according to the downscaled temperatures T following Robinson et al. (2010), where $SF = f(T)PP_{int}$, with the solid fraction of the monthly mean precipitation, f(T) following a sine
function from 1 to 0 between threshold temperatures $T_{snowy} = -7\,°\mathrm{C}$ and $T_{rainy} = 7\,°\mathrm{C}$, below and above these thresholds all precipitation is considered to be snow and rain. respectively.

**Statistical downscaling of radiative fluxes for fair and cloudy conditions**

We apply equation 8 to downscale longwave radiation ($LW$) for distinct atmospheric emissivities $\epsilon_{fair}$ and $\epsilon_{cloudy}$ for fair and cloudy condiotions so that

$$LW = CC \cdot LW(\epsilon_{cloudy}, T) + (1 - CC)LW(\epsilon_{fair}, T). \tag{10}$$

Similarly, insolation is a function of top of the atmosphere (TOA) insolation

$$SW^{\downarrow} = CC\tau_{cloudy}\widehat{SW} + (1 - CC)\tau_{fair}\widehat{SW} \tag{11}$$

with atmospheric transmissivities $\tau_{cloudy}$ and $\tau_{fair}$ for fair and cloudy conditions.

To separate the involved atmospheric parameters for cloudy and fair conditions, we need to introduce two additional as-
sumptions which are based on an analysis of PROMICE automatic weather station data (Ahlstrom et al., 2008). Specifically we analyze daily radiation, cloud cover, and air temperature observations from 17 stations, which cover up to 11 years (Fig. 2).

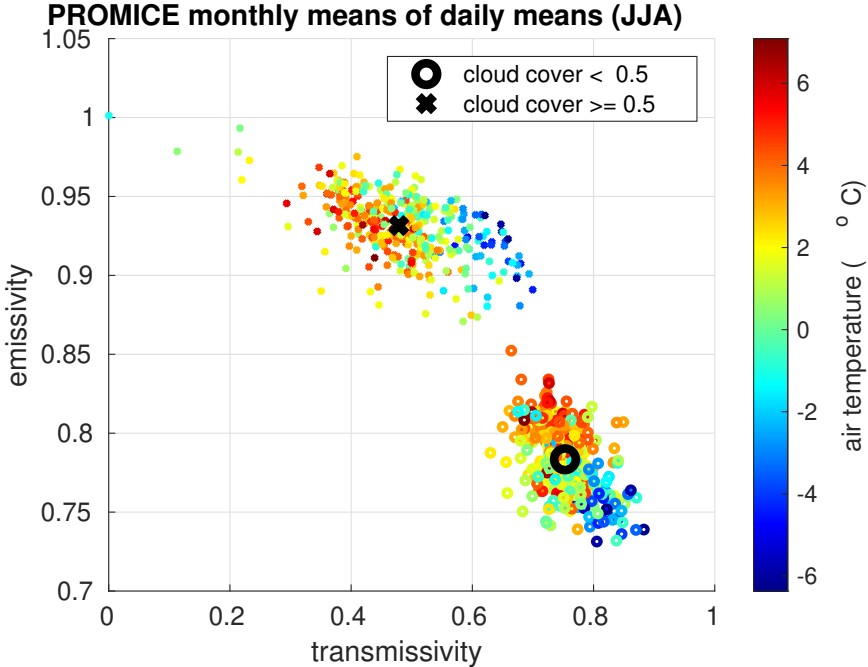

**Figure 2.** Monthly mean emissivities versus transmissivities for fair and cloudy conditions, $\overline{\epsilon}_{fair}, \overline{\epsilon}_{cloudy}, \overline{\tau}_{fair}$ and $\overline{\tau}_{cloudy}$ of all summer months as calculated from up to 11 years of daily observations from 17 PROMICE weather station. Every symbol represents the respective parameters as diagnosed for one individual month at one station. Colours reflect the respective air temperature measurements. Black symbols represent the respective dataset means.

We classify all summer days (June to August) with cloud cover $\geq 50\%$ as "cloudy" and otherwise as "fair and calculate monthly mean $\epsilon_{fair}, \epsilon_{cloudy}, \tau_{fair}$ and $\tau_{cloudy}$.

Under fair conditions atmospheric transmissivity $\tau_{fair}$ is relatively well constrained (Fig. 2). Therefore, we set $\tau_{fair} = 0.75$ to diagnose $SW\downarrow_{fair} = \tau_{fair}\widehat{SW}$ and $SW\downarrow_{cloudy} = \frac{1}{CC}(SW\downarrow - (1 - CC)SW\downarrow_{fair})$ from equation 11. To avoid numeric problems, we only apply this separation, if monthly cloud cover is in the range of [0.1 0.9] and otherwise use unseparated $SW\downarrow$ and $\epsilon$ to calculate the energy balance $Q$, accounting (not accounting) for the diurnal melt period during the entire month, if $CC < 0.1$ ($CC < 0.9$), respectively.

We constrain atmospheric emissivities by assuming that

$$\epsilon_{cloudy} = \epsilon_{fair} + \Delta\epsilon. \tag{12}$$

This is in line with parameterizations which assume that atmospheric emissivity independently depends on greenhouse gas concentration (which is primarily water vapor) and cloud cover (König-Langlo and Augstein, 1994, e.g.). The difference between $\epsilon_{fair}$ and $\epsilon_{cloudy}$ is assumed to be constant and both values are equally affected by greenhouse gasses. For a given





$\Delta\epsilon$, we can determine

$$\begin{aligned} \epsilon_{fair} &= \epsilon_a - CC\,\Delta\epsilon \\ \epsilon_{cloudy} &= \epsilon_a - (1-CC)\Delta\epsilon. \end{aligned} \tag{13}$$

According to König-Langlo and Augstein (1994); Sedlar and Hock (2009) the emissivity difference will be $\Delta\epsilon \approx 0.21$ if cloudy and fair conditions correspond to a cloud cover of $100\%$ and $0\%$, respectively. This value is not realistic, because partially cloud covered days occur frequently (Fig. 1), and emissivity and cloud cover are not linearly related. Instead Fig. 2 indicates that $\Delta\epsilon \approx 0.155$, which is the value we have chosen in all applications.

**Positive degree days**

To parameterize the mean temperature of the diurnal melt period $T_{MP}$ from monthly mean temperatures we resort to positive-degree-days, $PDD$, where $PDD$ is the temporal integral of near-surface temperatures $T$ exceeding the melting point. As in Krebs-Kanzow et al. (2018b) we use $T_{MP} = PDD_{\sigma=3.5}$, with $PDD_{\sigma=3.5}$ approximated as in Calov and Greve (2005) from monthly mean near-surface temperature $\overline{T}_a$ and a constant standard deviation of $\sigma = 3.5\,^\circ\mathrm{C}$

## 2.5 Initialization and forward integration

For a transient simulation, we initialize the model with no initial snow cover ($SNH = 0$) and, for Northern Hemisphere applications, start the integration with October, the beginning of the hydrological year. After December, we continue the integration (re-)using the forcing of the first year for two 12-month cycles. The first 15 months are considered to be a spin-up, the following second full cycle is the first year of the actual simulation. At the end of each month, snow height is updated according to its surface mass balance (section 2.7). After September we additionally subtract the snow height of previous year's September, which corresponds to the assumption that snow which is by then older than a year will transform into ice. On the Southern Hemisphere, the integration should start in April and snow should transform to ice by the end of March.

## 2.6 The albedo scheme

Surface melt decreases the albedo of snow and ice and at the same time a lowered albedo intensifies surface melt. This strong positive feedback is a particularly crucial mechanism accelerating the recent mass loss of the GrIS (Box et al., 2012). The albedo of ice and snow is thus a critical parameter in any surface mass balance estimate which is based upon the balance of radiative and turbulent energy fluxes. In dEBM albedo of new snow, $A_{NewSnow}$, dry snow, $A_{DrySnow}$ and albedo of wet snow or ice, $A_{WetSnow}$ are distinguished. To determine the local surface type for a given month we preliminary calculate for these $ME(A_{\mathrm{SurfaceType}})$ and $RZ_{pot}(A_{\mathrm{SurfaceType}})$ as a function of albedo for each surface type. The local albedo is then determined by testing a seqence of logical conditions which are illustrated as a decision tree in Fig. 3. Under cloud-covered conditions we assume that albedo increases by 0.05 following the work of Willeit and Ganopolski (2018).



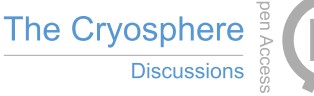

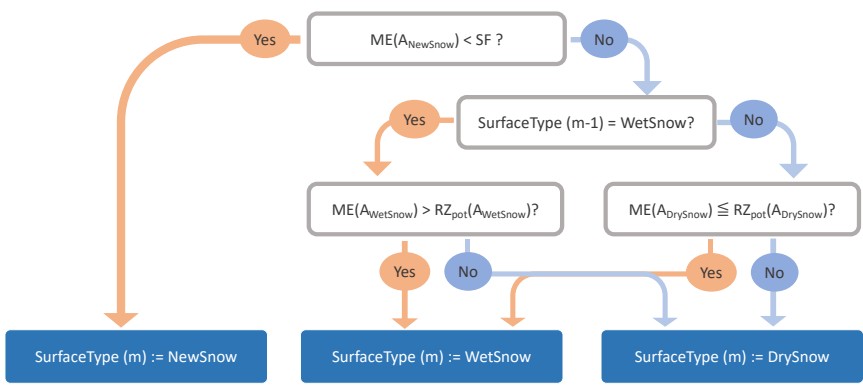

**Figure 3.** Schematic of the algorithm which selects the surface type (NewSnow, DrySnow or WetSnow) for each grid point and month m.

## 2.7 Snow height

At the end of every month m we update the height of the surface snow layer according to

$$SNH(m) = \max(0, SNH(m-1) + \frac{\Delta t}{\rho}(SF - ME + RZ)). \tag{14}$$

It is important to note that, between months, water cannot be stored within the snow column. That part of the monthly produced
melt water which does not refreeze within the same month will be removed from the snow column and will be added to the
runoff. At the end of September we suppose, that snow which is older than a year has been transformed to ice and accordingly
reset snow height to

$$SNH(m) = SNH(m) - SNH(m-12).$$

## 3 Parameter selection and evaluation based on observations

### 3.1 Experimental design

The choice of the albedo substantially influences the sensitivity of all SMB schemes which are based on the surface energy
balance. The dEBM distinguishes three surface types, new snow, dry snow and wet snow, each associated with a distinct albedo.
Similar surface types can be distinguished in observarion. In field measurements on the western GrIS, Knap and Oerlemans
(1996) observe albedos between 0.85 and 0.75 at higher altitudes and after the end of the melt season. Similarly Aoki et al.
(2003) find that albedo of dry snow ranges between $\approx 0.85$ for fresh snow and 0.75 for aged snow. During the melt season
and near the ice edge, Knap and Oerlemans (1996) find a wide range of albedos for different surface types ranging from





albedos around 0.45 for ice with ponds of surface water to mean albedos around 0.65 for superimposed ice (fragmented ice with an angular structure). On larger spatial scales and averaged over multiple days, however, areas of wet snow and ice typically exhibit albedos between 0.5 and 0.58 (Bøggild et al., 2010; Alexander et al., 2014; Riihelä et al., 2019). We conduct a series of calibration experiments with different parameter combinations for $A_{NewSnow}$, $A_{DrySnow}$, $A_{WetSnow}$ together with

the residual heat flux R (in equation 6). We vary $A_{ns}$ within $[0.84, 0.845, 0.85]$, $A_{ds}$ within $[0.68, 0.69, 0.70, ..., 0.78]$, $A_{ws}$ within $[0.53, 0.54, 0.55, 0.56, 0.57]$ and R within $[-2, -1, 0]$. These calibration experiments adapt the experimental design of (Fettweis et al., 2020) and simulate the 1980-2016 SMB of the GrIS using monthly ERA-Interim forcing (Dee et al., 2011) interpolated or downscaled by dEBM to the 1km ISMIP6 grid (Nowicki et al., 2016). We evaluate these experiments based on two independent observational data set. In the following we refer to these data sets as *local observations* and *integral*

*observations*.

**Local observations**

We evaluate the calibration experiments based on local SMB measurements from Machguth et al. (2016) which are distributed around the ice sheet's margins and which provide integral SMBs over periods between months and multiple years. For each calibration experiment we bilinearly interpolate the simulated SMB of the four nearest grid cells of the ISMIP6 grid to the

coordinates of the measurements and integrate simulated SMB over the respective observation period. Where observations do not cover full months the respective simulated monthly mean values contribute proportionally. We do not include observations which are outside of the ISMIP6 ice mask, which are not completely covered by the 1980-2016 period or which cover less than three months, which leaves 1252 local observations which primarily allow to assess the skill of the model to reproduce spatial characteristics of the SMB.

**Integral observations**

Also, we compare the simulated SMB to the 2003-2016 annual integral Greenland SMB derived from the sum of GRACE mass balance measurements $\frac{\Delta M}{\Delta t}$ (Sasgen et al., 2012, 2020) and interpolated monthly estimates of solid ice discharge D from Mankoff et al. (2019), assuming that $SMB = \frac{\Delta M}{\Delta t} + D$. Using the integral observations we calculate annual SMB from October 2003 to September 2015 based on hydrologic years which start in October, which then provides the basis to assess the skill of

the model to reproduce the integral SMB and its interannual variability.

For the evaluation of the dEBM scheme we take into account that the precipitation forcing is possibly biased: snow accumulation at the slopes of the ice sheet may be underestimated or low accumulation rates in the interior may be relatively inaccurate in the ERA-Interim reanalysis. Note that we do not optimize any parameters which affect snow accumulation and errors in

the forcing data may influence the calibration. Furthermore GRACE observations include regions of Greenland with seasonal snow cover and ice caps, which are not part of the ISMIP6 main ice sheet domain. We here assume that both, errors in the in-



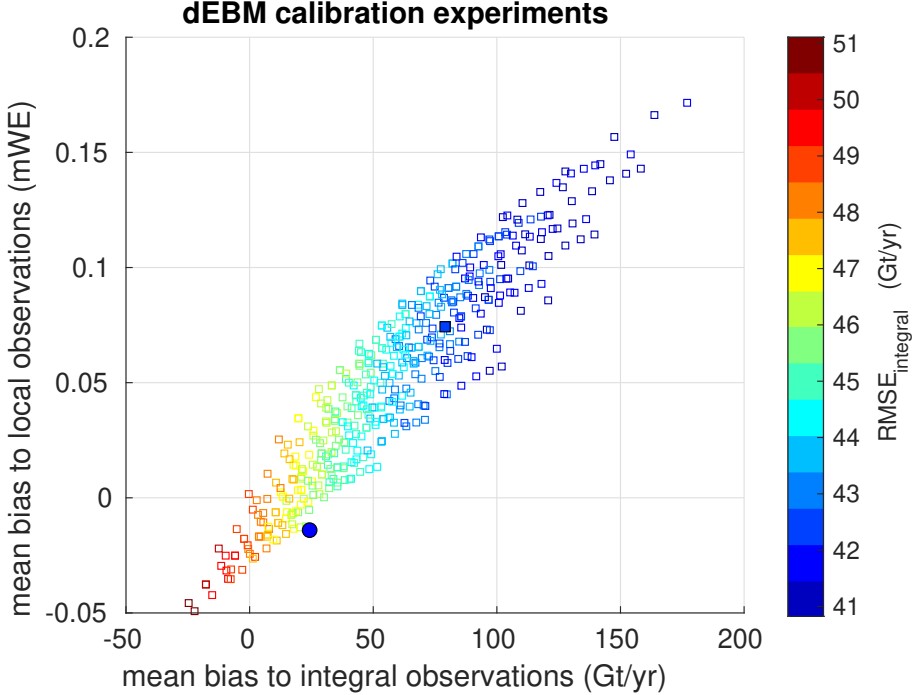

**Figure 4.** Mean bias of calibration experiments to integral observations as a function of the mean bias to local observations. The colour represents the root mean square error of annual variations of the calibration experiments relative to the annual variations in the integral observations after the respective mean bias was removed. The experiment with the parameter combination which was selected for all following experiments is highlighted as a filled square with black borders. Also shown is experiment $dEBM_{MAR,ERA}$ as a solid circle, which uses the same selected parameter combination but different forcing (see section 4).

terpolated ERA-Interim precipitation and the inconsistency of domains primarily affect the multi-year mean SMB of the GrIS but not so much its spatial or interannual variations. For this reason we separately evaluate the mean SMB and the variation around the mean with respect to the integral and local observations.

### 3.2 Analysis of the calibration experiments

5   We find that agreement of simulated mean SMB with observations is consistent between local and integral observations (Fig. 4), (low bias to local observations is also associated with low bias to integral observations), which indicates that the systematic bias over the entire period and domain is small in both datasets. Furthermore, we find multiple parameter combinations which yield reasonable agreement both to the temporal variations in the integral observations and to the spatial structure in the local observations (Fig. 5). Good agreement to variations in both datasets ($RMSE_{integral} < 43$ and $RMSE_{local} < 0.557$)

10   is associated with a bias of approximately -80Gt/year to the mean integral observations (Fig. 5) and a bias of approximately -0.07 mWE with respect to mean local observations. Closer inspection of the parameter combinations (not shown) which yield

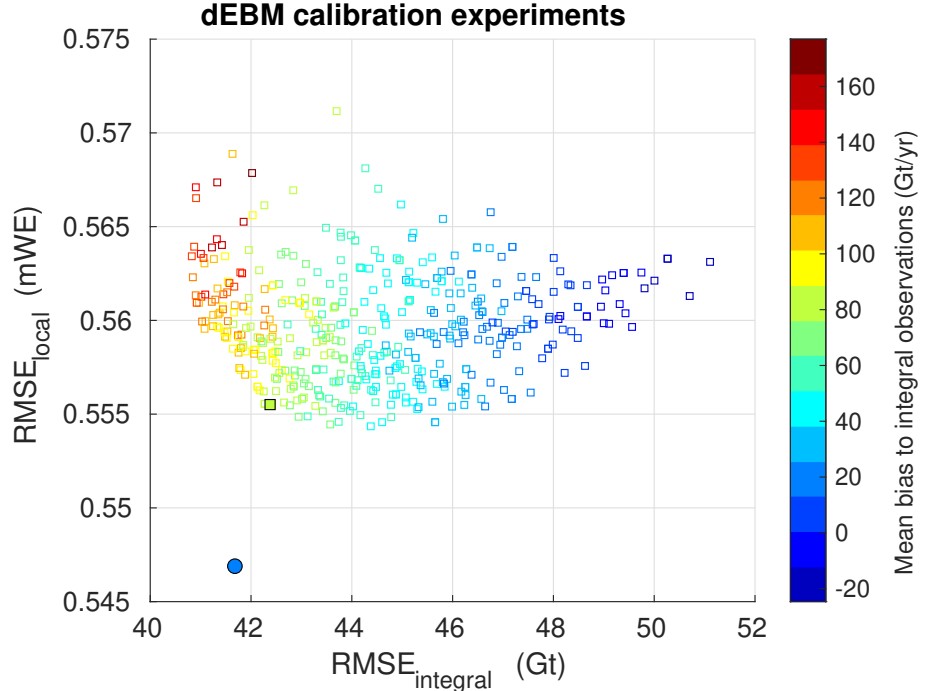

**Figure 5.** Root mean square error of the calibration experiments to temporal variations in the integral observation as a function of the root mean square error of calibration experiments to local observations. The mean bias between observations and calibration experiments has been removed before root mean square errors were calculated. Colors represent the mean bias of the calibration experiments to integral observations. The experiment with the parameter combination which was selected for all following experiments is highlighted as a filled square with black borders. Also shown is experiment $dEBM_{MAR,ERA}$ as a solid circle, which uses the same selected parameter combination but different forcing (see section 4).

such good agreemeent reveals that combinations with $R = 0$, $A_{NewSnow} = 0.845$ and $A_{WetSnow} = 0.55$ or $A_{WetSnow} = 0.56$ provide a generally good skill. Varying $A_{DrySnow}$ in the range of [0.65 0.75] mostly influences whether agreement to local or integral observations is better.

Based on the calibration experiments, we choose the parameter combination of $R = 0$, $A_{NewSnow} = 0.845$ and $A_{DrySnow} = $
5   0.73 and $A_{WetSnow} = 0.55$ for all following experiments. Using this combination together with the ERA-Interim forcing in experiment $dEBM_{ERA}$ yields a good agreement with both the local and the integral observations.



## 4 Evaluation based on the regional climate model MAR

### 4.1 Experimental design

To compare dEBM to the regional model MAR we conduct an experiment which follows the design of the calibration but uses a modified precipitation forcing. Experiment $dEBM_{MAR,ERA}$ uses dynamically downscaled snow and rainfall from an ERA-Interim forced simulation with the regional climate model MAR (experiment $MAR_{ERA}$, Fettweis et al., 2020), all other forcing fields are identical to the forcing used for the calibration experiments. The experiment $MAR_{ERA}$ here also serves as a reference for comparison. The MAR simulation $MAR_{ERA}$ was conducted in the framework of the Greenland Surface Mass Balance Intercomparison Project (GrSMBMIP) on an equidistant 15km grid and was forced with 6-hourly ERA-Interim data at its lateral boundaries. This simulation was found to be in particularly good agreement with observations (Fettweis et al., 2020).

### 4.2 Evaluation of experiment $dEBM_{MAR,ERA}$

Replacing the precipitation forcing by $MAR_{ERA}$ precipitation considerably improves the agreement with local observations. Furthermore, experiment $dEBM_{MAR,ERA}$ exhibits a smaller mean bias to observations (Fig. 4 and Fig. 5) which supports our earlier hypothesis that the mean bias in experiment $dEBM_{ERA}$ may be related to a systematically biased precipitation in the coarse resolution ERA-Interim forcing. Experiments $dEBM_{MAR,ERA}$ and $MAR_{ERA}$ generally agree with respect to the evolution of integral, annual SMB (Fig. 6) with a root mean square error of $27\,\mathrm{Gt}$. In comparisson to the seasonal cycle of MAR, dEBM underestimates/overestimates early/late summer SMB which indicates that dEBM fails to accurately reproduce onset and end of the annual melt season due to its monthly time step (Fig. 7). Integrated over the year these seasonal biasses mostly cancel out.

We now evaluate the spatial representation of components of the SMB by comparing experiment $dEBM_{MAR,ERA}$ to the $MAR_{ERA}$ simulation for the period 1980 to 1999. By design the two simulations are identical in snow accumulation while variables which influence the meltwater runoff (i.e. temperatures, radiation and cloud cover) are dynamically consistent but not identical to the respective forcings used in $dEBM_{MAR,ERA}$. The presented MAR output has been interpolated or, in case of air temperature and SMB, downscaled from its native 15km resolution to the 1km ISMIP6 grid. The temperature forcing of dEBM has been downscaled from ERA-Interim fields using a fixed lapse rate of $7\,\mathrm{K\,km^{-1}}$ and generally exhibits a similar spatial structure. In comparison to MAR summer temperatures we observe a large-scale warm bias over high elevation North Greenland and mostly negative anomalies along the Eastern margins of the ice sheets which can exceed $5\,°\mathrm{C}$ around the complex East Greenland fjord systems around the Scoresby Sound. Large-scale patterns are inherent differences between ERA-Interim and MAR while local difference, especially at the coasts, may partly be a result of the relatively crude lapse rate correction of the dEBM forcing (Fig. 8).

Surface melt rates largely agree in the ablation zones of West and Southeast Greenland (Fig. 9). Considerably weaker dEBM melt rates in the region of the Scoresby Sound can be attributed to the lower temperatures in the dEBM forcing, while lower melt rates at the southern tip of Greenland are not associated with respective differences in the temperature forcing. Stronger

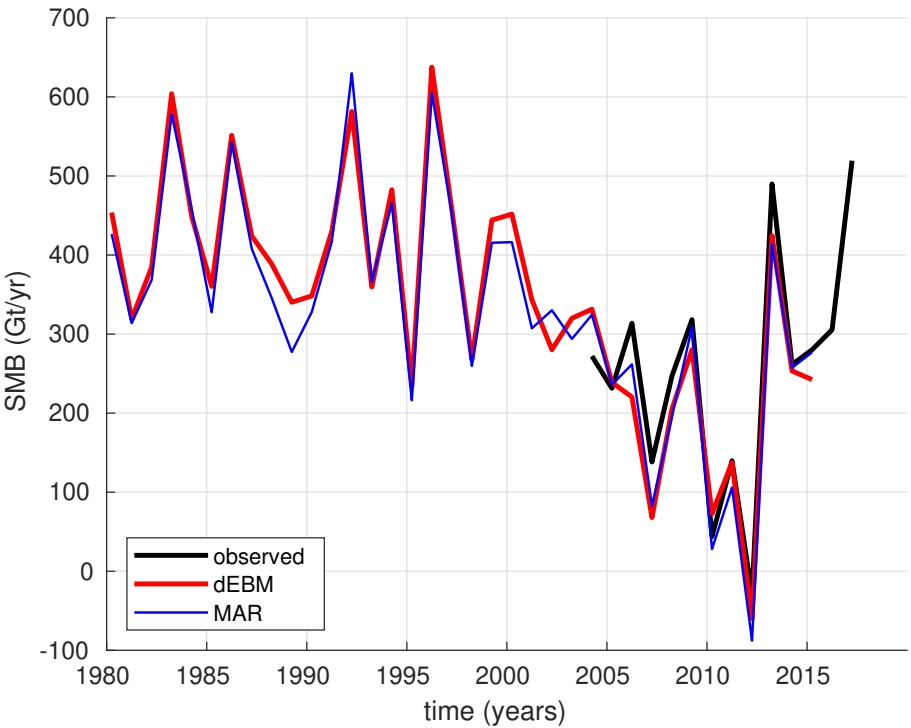

**Figure 6.** Annual mean integrated SMB in $\mathrm{Gt\,yr}^{-1}$ of the Greenland Ice Sheet as derived from integral observations (black), and as simulated by MAR (blue) and dEBM (red)

melting at the ice sheet's fringes is particularly visible in North Greenland and at the Southeastern cost which is also not always associated with warmer temperatures. Differences which are not explained by different temperature forcing may have a multitude of reasons: the simplicity of the dEBM albedo scheme, unresolved sub-monthly variability or the (neglected) effect of humidity and high wind speed on turbulent heat fluxes which will be important at coastal locations. Finally dEBM seems

5   to underestimate melting systematically at the upper boundary of the ablation zone. This is likely related to unrepresented sub-monthly temperature variability, as temperatures exhibit stronger variability at high elevations (Fausto et al., 2011). The weaker melting at higher elevations is in part compensated by refreezing which is generally weaker in dEBM than in MAR and especially in the higher parts of the ablation zone.

In total we find a good agreement between simulated SMB from dEBM and MAR with differences being mostly restricted

10  to narrow regions at the coast (Fig. 10).

The simulated albedo is closely linked to the simulated SMB. In the interior of the ice sheet, albedos simulated by dEBM are generally up to 0.05 higher than MAR albedos. Outside of the ablation zone MAR simulates a gradual transition towards higher albedos while dEBM uses always the new snow albedo of $A_{NewSnow} = 0.845$ as soon as melt rates fail to exceed snowfall. If we use MAR as a reference, we find that within the ablation zone, dEBM seems to underestimate albedos in regions with high



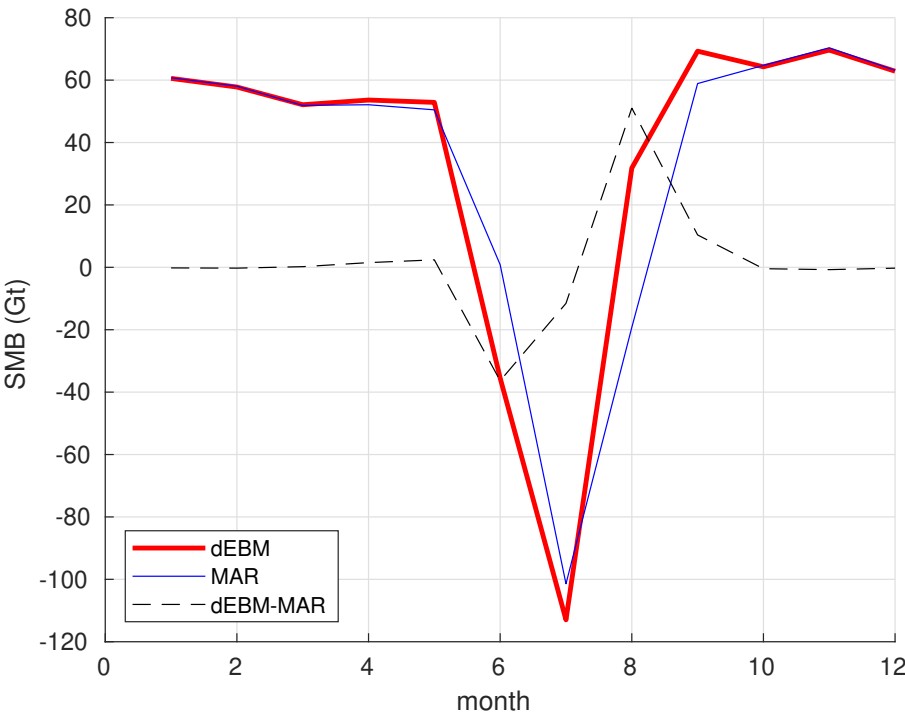

**Figure 7.** 1980-1999 multiyear monthly means of the GrIS SMB in $\mathrm{Gt}$ as simulated by MAR (blue) and dEBM (red) and their difference (dashed black)

accumulation rates (Southwest and South East) while albedo is mostly overestimated in the North. Remarkably, higher/lower albedo in the ablation zone is not necessarily associated with accordingly higher/lower SMB or vice versa.



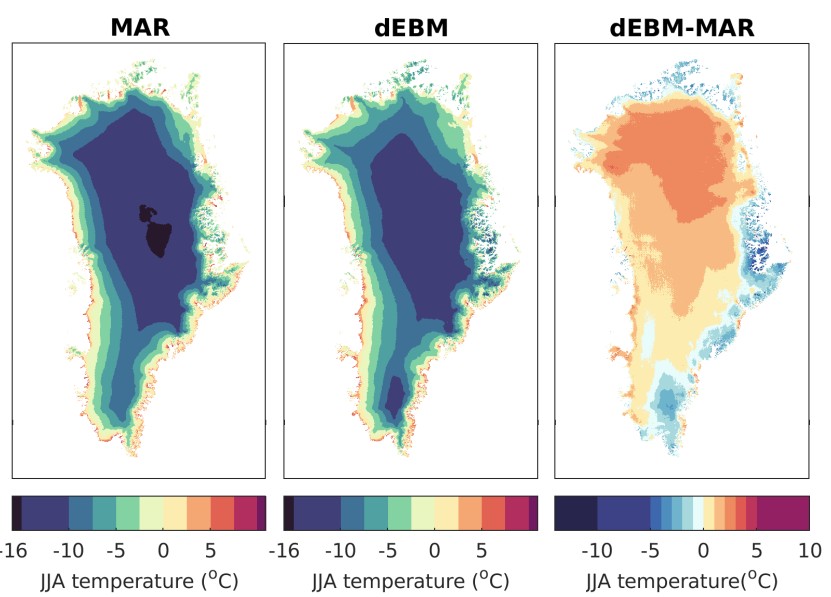

**Figure 8.** Comparison of multi-year (1980-1999) mean summer near-surface temperature from experiment $MAR_{ERA}$ (left), $dEBM_{MAR,ERA}$ (center) and differences between $dEBM_{MAR,ERA}$ and $MAR_{ERA}$ (right).



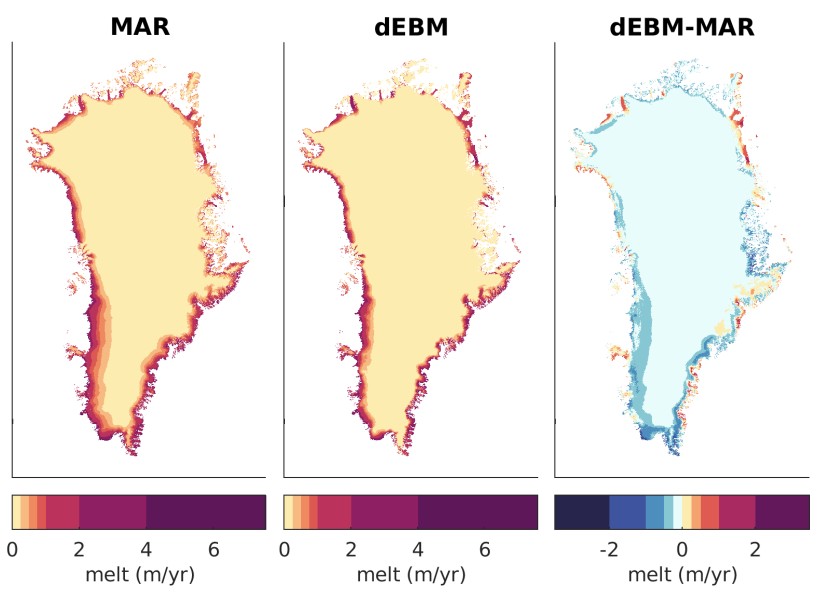

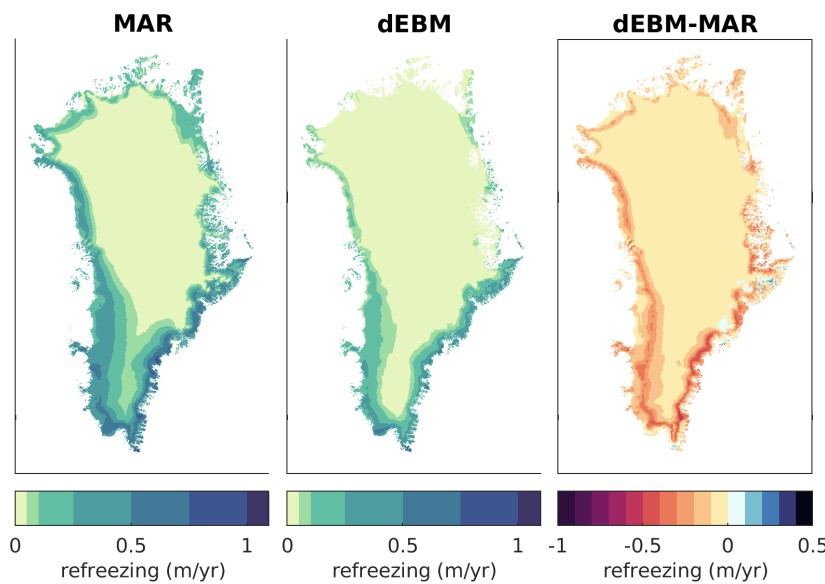

**Figure 9.** Comparison of multi-year (1980-1999) mean melt rates (upper row) and refreezing rates (lower row) from experiment $MAR_{ERA}$ (left), $dEBM_{MAR,ERA}$ (center) and differences between $dEBM_{MAR,ERA}$ and $MAR_{ERA}$ (right).



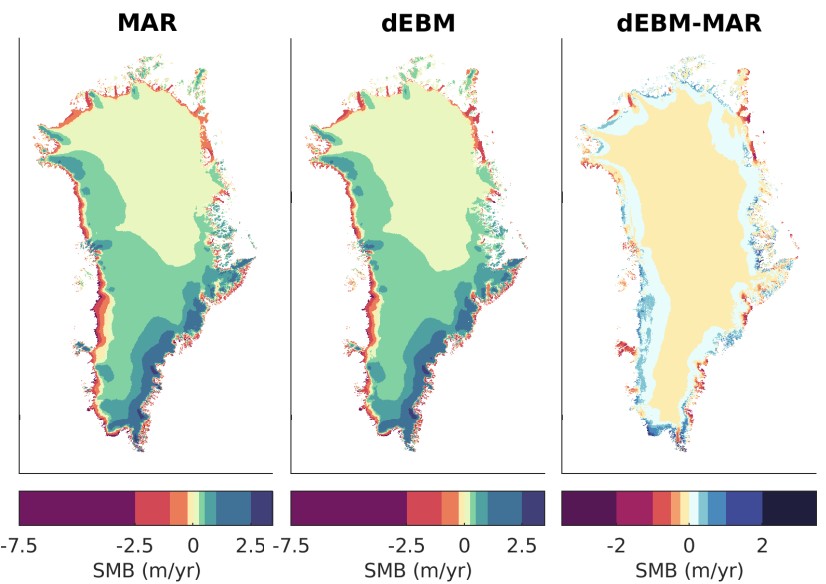

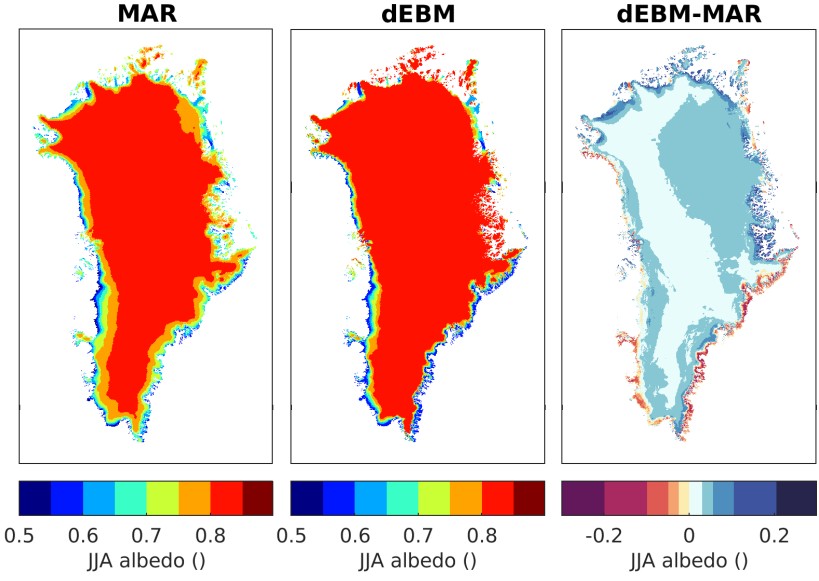

**Figure 10.** Comparison of multi-year (1980-1999) mean surface mass balance (upper row) and summer albedo (lower row) from experiment $MAR_{ERA}$ (left), $dEBM_{MAR,ERA}$ (center) and differences between $dEBM_{MAR,ERA}$ and $MAR_{ERA}$ (right).



## 5 Sensitivity of the SMB to climate

### 5.1 Experimental design

We use dEBM to study the SMB of the Greenland Ice Sheet in a warm climate period of the past and in the warming climate of a future climate scenario. Both simulations have been conducted with the AWI Earth System Model, AWI-ESM (Sidorenko

5 et al., 2015) and both experiments are using an invariant present day ice sheet geometry as boundary conditions.

**Mid Holocene simulation H6K**: Due to a stronger than present axial tilt of the Earth (obliquity) the Mid Holocene (6000 years before present) was characterized by intensified summer insolation and consequently $2\,^{\circ}C\,to\,3\,^{\circ}C$ warmer summer temperatures over Greenland (Dahl-Jensen et al., 1998). The experiment *H6K* uses 200 years of monthly mean climate forcing from an equillibrated Mid Holocene simulation. The Mid Holocene simulation has been conducted using modified orbital

10 parameters and greenhouse gas concentration following the PMIP protocols as defined in Otto-Bliesner et al. (2017).

**1850 to 2099 simulation Industrial**: Experiment *Industrial* uses 250 years of monthly forcing from an experiment with changing boundary conditions which is a combination of a historical simulation from 1850 to 2005 followed by a future projection forced according to a high emission scenario (following the "representative concentration pathway" RCP8.5, as described in Taylor et al. (2012)).

15 In the following we use the years 1850 to 1899 of experiment *Industrial* as a reference period ("PI" hereafter) for both experiments which here serves as a surrogate for the preindustrial period. The AWI-ESM forcing is here downscaled to the equidistant 5km grid.



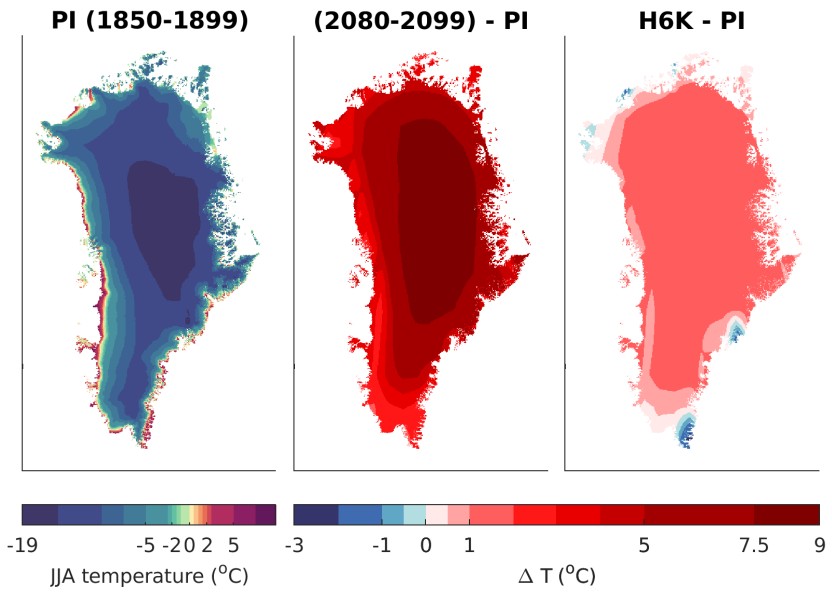

**Figure 11.** Mean summer 2m temperature of the PI period (years 1850 to 1899) in experiment *Industrial* (left) and anomalies of summer mean 2m temperarature with respect to PI of the *Industrial* 2080 to 2099 period (center) and mean Mid Holocene (H6K) summer 2m temperatures (right)



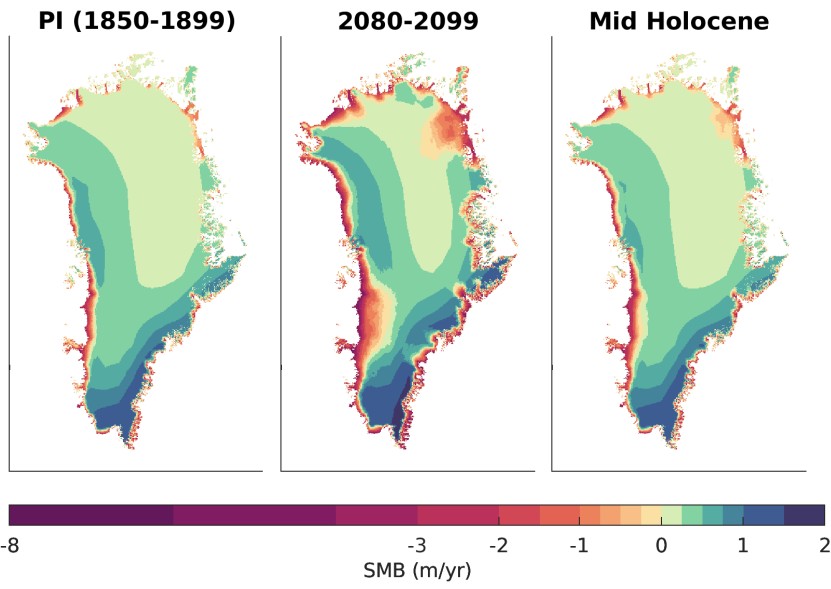

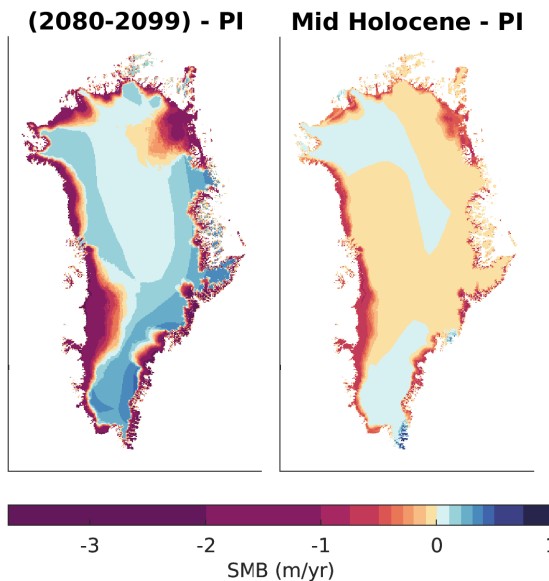

**Figure 12.** Upper row: Mean SMB of experiment *Industrial* during the PI period (years 1850 to 1899, left), during the 2080 to 2099 period experiment *Industrial* (center) and Mean SMB of experiment *H6k*. Lower row: SMB anomaly with respect to the PI period for the years 2080 to 2099 *Industrial* (left) and for experiment H6K



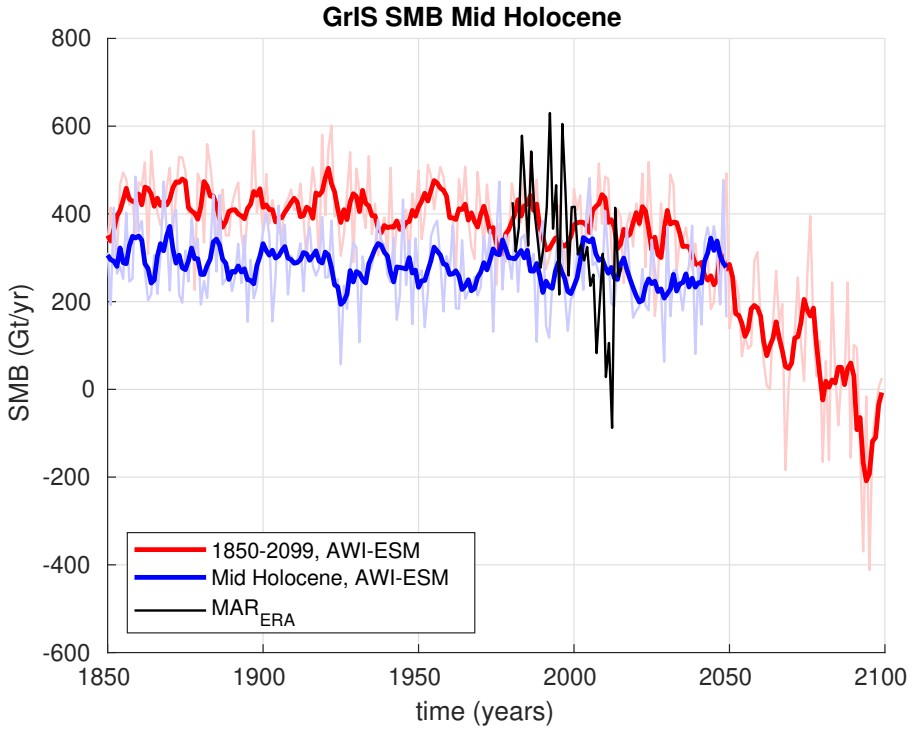

**Figure 13.** GrIS SMB timeseries from experiment *Industrial* (red), experiment {H6K (blue) and experiment $MAR_{ERA}$ (black, section 4). Fine lines show yearly accumulated values, bold lines represent respective 5-year moving means.

## 5.2 Experiments H6K and Industrial

Compared to the PI period of experiment *Industrial*, the climate of experiment *H6K* is characterized by stronger insolation and higher air temperatures over Greenland in summer (Fig. 11). In the ablation zone summer air temperatures exceed PI temperatures by approximately $0.5\,°C$ to $1\,°C$ (Fig. 11, which is somewhat lower than reconstructed temperatures (Dahl-Jensen et al., 1998). The experiment *Industrial* exhibits a strong warming in the 21st century with summer air temperatures at the ice sheet's margins rising by $3\,°C$ to $5\,°C$ above late 19th centuries values (Fig. 11).

In response to the warmer climate of experiment *H6K*, dEBM simulates intensified melting and generally slightly extended ablation areas (Fig. 12), which in total decreases the mean SMB of the entire ice sheet by more than $100\,\mathrm{Gt}$ (Fig. 13). The transient climate of experiment *Industrial* yields only a minor trend in SMB throughout the 20th century and starts to decrease substantially in the first half of the 21st century. By the end of the simulation SMB has decreased by more than $500\,\mathrm{Gt}$ and the total SMB of the GrIS has changed its sign to negative. Especially in the West and Northeast the ablation zone is no longer restricted to the margins but extends to the interior ice sheet (Fig. 12). The intensified melting is to some part compensated by higher accumulation rates. Simulated SMB around the end of the 20th century agrees well with the MAR simulation. The climate model however does not reproduce the extreme Greenland blocking in the 2005–2015 period, which is a common



problem in global climate models (Hanna et al., 2018). Accordingly the interannual variations in SMB of recent decades is underestimated and the simulated negative trend in SMB may be delayed.

## 5.3 Analysis of the temperature-melt relation

Local observations from Greenland reveal a linear relation between positive-degree days (PDD) and surface melt scaling with so-called degree-day factors. This linear relationship is the basis of many empirical models (e.g. Reeh, 1991). For ice sheet applications degree-day factors are commonly chosen to be $DDF_{ice} \approx 8\,\mathrm{mm\,K^{-1}\,d}$ and $DDF_{snow} \approx 3\,\mathrm{mm\,K^{-1}\,d}$ for snow and ice respectively (Lefebre et al., 2002; Huybrechts et al., 1991). We now investigate the sensitivity of this relation between temperature and melt under different climates. For this purpose we separately integrate total simulated annual melt rates, positive degree days and the temperature independent terms of the surface energy balance (Eq. 5) $(1-A)SW\downarrow +b$ over two surface type. We here do not distinguish between snow and ice but classify all local monthly melt rates $ME$ of a given year into two subsets $ME_{ws}$ where the $SurfaceType = WetSnow$ and $ME_{ns,ds}$ where $SurfaceType \neq WetSnow$. Analoguesly we analyze positive degree-days and the temperature independent terms of Eq. 5 $(1-A)SW\downarrow +b$ in respective subsets of yearly output. Based on our dEBM simulations we then infer degree-day factors $DDF_{ws}$ and $DDF_{ns,ds}$ as in Krebs-Kanzow et al. (2018a) according to

$$
\begin{aligned}
DDF_{ws} &= \frac{\sum ME_{ws}}{\sum PDD_{ws}} \\
DDF_{ns,ds} &= \frac{\sum ME_{ns,ds}}{\sum PDD_{ns,ds}}
\end{aligned}
\tag{15}
$$

which represent an annual mean of all local degrre-day factors weighted by the melt rate (Fig. 14). Despite the inconsistency in the classification of surface types, we find a general agreement with the empirical parameters with $\overline{DDF}_{ws} = 8.7\,\mathrm{mm\,K^{-1}\,d}$ and $\overline{DDF}_{ns,ds} = 2.1\,\mathrm{mm\,K^{-1}\,d}$ averaged over the 1850 to 1999 period. Both $DDF_{ws}$ and $DDF_{ns,ds}$ are especially sensitive to the *H6K* background climate with mean degree day factors $\overline{DDF}_{ws} = 9.8\,\mathrm{mm\,K^{-1}\,d}$ and $\overline{DDF}_{ns,ds} = 3.1\,\mathrm{mm\,K^{-1}\,d}$. Sensitivity to the warming climate of the 21st century is less pronounced with both degree-day factors increasing by $\approx 0.3\,\mathrm{mm\,K^{-1}\,d}$ towards the end of experiment *Industrial*. Comparison with the temperature independent terms of the surface energy balance (Eq. 5) indicates a linear relation between degree-day factors and temperature independent energy fluxes. The effect of shortwave radiation is in fact implicitly temperature dependent as surface albedo of glaciated surfaces usually decreases when air temperature exceeds melting point. This temperature dependence is also included in some albedo parameterizations (Bougamont et al., 2005; Aoki et al., 2003) and to some degree also represented in the dEBM by distinguishing three surface types.

## 6 Conclusion

The atmosphere influences the surface mass balance (SMB) of ice sheets on short temporal and small spatial scales, which induces long-term changes in continental ice volume in a changing climate. Usually, climate simulations that span more than a few centuries do not provide the required resolution to reliably predict the SMB, which implicates the necessity to down-



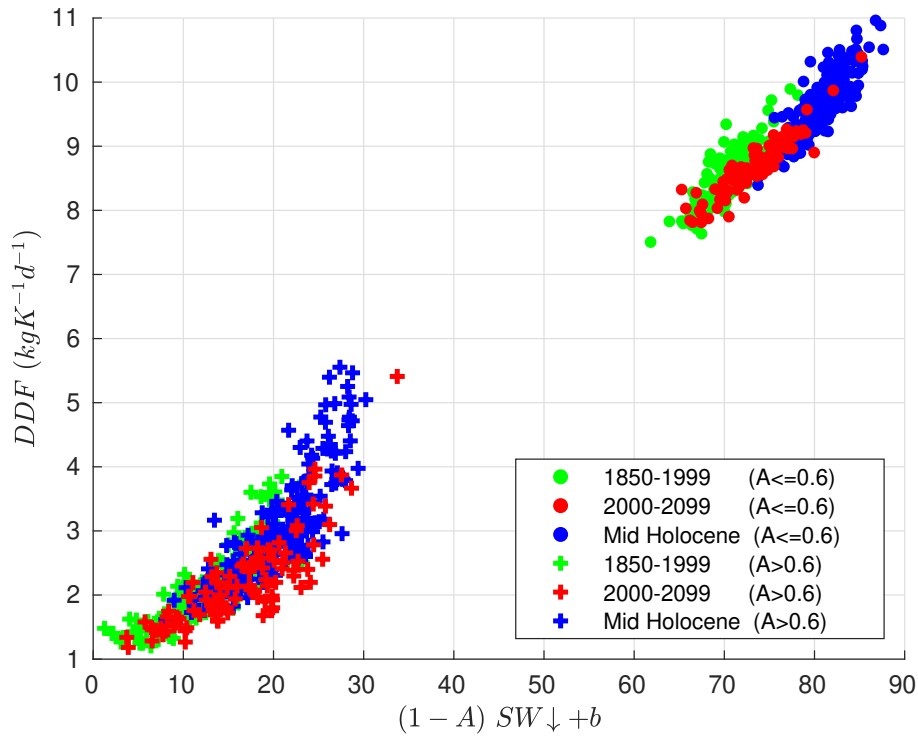

**Figure 14.** Degree-day factors diagnosed from dEBM for years 1850-1999 of the *Industrial* simulation (green), for years 2000-2099 of the *Industrial* simulation (red) for the 200 years of experiment *H6K* (blue) as a function of the temperature independent terms in Eq. 5. Each circle (cross) represents a domain-wide annual mean of all monthly values for which SurfaceType is WetSnow (SurfaceType is not WetSnow) weighted by melt rate

scale climate forcing on long timescales. Here, we introduce the diurnal Energy Balance Model, dEBM, an SMB model of intermediate complexity. The dEBM is particularly suitable for Earth System modeling on multi-millennial time scales as model parameters are sufficiently general to remain applicable if atmospheric greenhouse gas concentration, or the seasonal and diurnal cycle change. The central concept of this model is the temporal downscaling that accounts for both sub-monthly

5 variations in cloud cover and the diurnal melt-freeze cycle (Krebs-Kanzow et al., 2018b). This approach allows us to calculate SMB from monthly forcing with a monthly time-step which reduces the computational cost substantially and provides for an uncomplicated interface.

The model is physically plausible, as optimal parameters, calibrated to SMB observations from the GrIS, remain well within observational constraints.

10 The presented version agrees better with observations than an earlier version that already has demonstrated reasonable skill to simulate the SMB of the GrIS (GrSMBMIP, Fettweis et al., 2020). The main progress in this new dEBM version is that atmospheric emissivity is no longer parameterized but is now diagnosed from the atmospheric forcing. Since Zolles and Born (2019) have demonstrated a strong sensitivity, the explicit inclusion of longwave radiation improves the computed SMB for a





broad spectrum of climate stages ranging from glacial to future climate projections with strong radiative forcing. The dEBM compares well with simulated SMB from the complex regional climate model MAR (Fettweis et al., 2020) and exhibits an overall good agreement with local and integral observations.

The dEBM does not downscale precipitation but interpolates precipitation forcing. A comparison of two dEBM simulations,
which only differ in precipitation forcing, either originating from the ERA-Interim (Dee et al., 2011) reanalysis or from the regional climate model MAR, indicates that the coarser resolution of the reanalysis data induces a systematic bias in precipitation over Greenland. Coarse-resolution precipitation forcing represents a general source of error but this is unlikely to affect the relative change of SMB in response to climate variations.

Furthermore, we have used dEBM in combination with two climate simulations from the global climate model AWI-ESM:
a simulation of the Mid Holocene warm period and a transient global warming scenario which covers the period 1850 to 2099. Both simulations exhibit warmer than present temperatures over the GrIS; the first one due to intensified summer insolation, the second one due to rising greenhouse gas concentration. In line with Plach et al. (2018), the sensitivity of surface melt to air temperature increases by more than $10\,\%$ in the Mid Holocene experiment. In contrast, the temperature-melt relation changes barely during the global warming scenario. Hence, empirical temperature based SMB methods like the commonly used PDD
method might be applicable for the next decades but are not reliable on millennial time scales or outside of Greenland.

Naturally, the reduced complexity and the monthly time step of our model implicate limitations. The comparison with MAR simulations reveals that the beginning and the end of the melt season is not truthfully simulated. In Greenland, under present-day climate these errors mostly cancel out but may also impair the representation of interannual variability. On orbital timescales, however, the melt season may be shorter or shifted in time, which may result in systematic errors over extended
periods. In principle, these errors could be reduced or assessed by testing different time-step schemes. Also owed to the monthly time step, dEBM may not reliably simulate the transition between dry and wet snow, where sub-monthly variability is usually strong (Fausto et al., 2011) and substantial surface melt may happen during short-lived warm spells. Furthermore, the model does not consider any processes within the snow column and relies on a simplistic albedo scheme, which may also impair the skill of the model near the upper boundary of the melt region or in regions with high accumulation rates. For these reasons the
dEBM may not be well suited for small scale applications, and it remains unclear whether this SMB model can be applied to individual glaciers or ice caps. In the context of long-term Earth system modeling, however, these shortcomings are probably outweighed by uncertainties in climate forcing and boundary conditions and do not substantially affect the sensitivity of the SMB to long-term climate change on the whole.

Nevertheless, some extensions and modifications might be considered, depending on the problem and the region of interest.
For the Antarctic ice sheet or the North American ice sheet, it might be advisable to add a sublimation scheme. In coastal regions, the constant temperature sensitivity of the turbulent heat flux might be replaced by a function of wind speed. Further improvements might focus on the heat flux to the ground, the representation of liquid water storage in the snow column, a shorter time step, and a downscaling algorithm for precipitation to reproduce better topographically steered precipitation.

As a natural next step we intend to test the dEBM in the framework of the coupled AWI-ESM Earth system model (Gierz
et al., 2020) to study glacial-interglacial timescales. Furthermore, dEBM can be used as a diagnostic for climate simulations



based on fixed ice sheet geometries or in conceptual studies that use synthetic forcing derived from climate models and observation as in Niu et al. (2019). Overall, dEBM captures the essential physics which drive SMB variations on long time scales. We envision this intermediate complexity model to be a low-cost alternative wherever dynamical downscaling with regional climate models is not feasible.

5  *Code availability.*  A FORTRAN version of the dEBM is available under https://github.com/ukrebska/dEBM/tree/update/Fortran-v1.0

*Competing interests.*  The authors declare that they have no competing interests





## Appendix A: Appendix

| name | variable | unit | type |
|---|---|---|---|
| $H$ | surface elevation | m | boundary condition |
| $PP$ | precipitation | $\mathrm{kg\,m^{-2}\,s^{-1}}$ | forcing |
| $T$ | near surface air temperature | K | forcing |
| $SW\downarrow$ | downward shortwave radiation at surface | $\mathrm{W\,m^{-2}}$ | forcing |
| $LW\downarrow$ | downward longwave radiation at surface | $\mathrm{W\,m^{-2}}$ | forcing |
| $\widehat{SW}$ | downward shortwave radiation at TOA | $\mathrm{W\,m^{-2}}$ | forcing |
| $Q$ | energy flux into the surface | $\mathrm{W\,m^{-2}}$ | calculated by the dEBM |
| $PDD$ | positive degree days per month | K | calculated by the dEBM |
| $SF$ | snow fall | $\mathrm{kg\,m^{-2}\,s^{-1}}$ | calculated by the dEBM |
| $RF$ | rain fall | $\mathrm{kg\,m^{-2}\,s^{-1}}$ | calculated by the dEBM |
| $ME$ | melt rate | $\mathrm{kg\,m^{-2}\,s^{-1}}$ | calculated by the dEBM |
| $RZ$ | refreezing rate | $\mathrm{kg\,m^{-2}\,s^{-1}}$ | calculated by the dEBM |
| $SMB$ | surface mass balance | $\mathrm{kg\,m^{-2}\,s^{-1}}$ | calculated by the dEBM |
| $RO$ | runoff | $\mathrm{kg\,m^{-2}\,s^{-1}}$ | calculated by the dEBM |
| $SNH$ | water equivalent snow height | kg | calculated by the dEBM |
| $\epsilon_a$ | atmospheric emissivity | | calculated by the dEBM |
| $\tau_a$ | atmospheric transmissivity | | calculated by the dEBM |
| $\Phi, q_\Phi, \frac{\Delta t_\Phi}{\Delta t}$ | characteristics of the diurnal melt period | | calculated by the dEBM |

**Table A1.** Variables used in the dEBM



| name | parameter | value | reference |
|------|-----------|-------|-----------|
| $\rho$ | density of liquid water | $1000\,\mathrm{kg\,m^{-3}}$ | |
| $L_f$ | latent heat of fusion | $3.34 \times 10^5\,\mathrm{J\,kg^{-1}}$ | |
| $T_0$ | melting temperature | $273.15\,\mathrm{K}$ | |
| $\sigma$ | Stefan-Boltzmann constant | $5.670\,51 \times 10^{-8}\,\mathrm{W\,m^{-2}\,K^{-4}}$ | |
| $\epsilon_i$ | longwave emissivity of ice | $0.98$ | Armstrong and Brun (2008) |
| $\gamma$ | slope lapse rate of temperature | $-7\,\mathrm{K\,m^{-1}}$ | typical value for ice sheet margins |
| $T_{snowy}, T_{rainy}$ | threshold temperature - rain/snow | $\pm 7\,^\circ\mathrm{C}$ | Robinson et al. (2010) |
| $T_{min}$ | threshold temperature - melting | $-6.5\,^\circ\mathrm{C}$ | Orvig (1954) |
| $\Delta\epsilon$ | emissivity difference between cloudy - fair days | $0.155$ | diagnosed from PROMICE data |
| $\tau_{fair}$ | transmissivity of fair days | $0.75$ | diagnosed from PROMICE data |
| $\beta$ | temperature sensitivity of turbulent heat fluxes | $10\,\mathrm{W\,m^{-2}\,K^{-1}}$ | Krebs-Kanzow et al. (2018b) |
| $R$ | unresolved heat flux | $1\,\mathrm{W\,m^{-2}}$ | calibrated parameter |
| $A_{NewSnow}$ | albedo of NewSnow | $0.845$ | calibrated parameter |
| $A_{DrySnow}$ | albedo of DrySnow | $0.73$ | calibrated parameter |
| $A_{WetSnow}$ | albedo of WetSnow | $0.55$ | calibrated parameter |

**Table A2.** Physical and empirical parameters used in this paper



*Acknowledgements.* U. Krebs-Kanzow is funded by the Helmholtz Climate Initiative REKLIM (Regional Climate Change), a joint research project of the Helmholtz Association of German research centres. P. Gierz is funded by the German Ministry of Education and Research (BMBF) German Climate Modeling Initiative PalMod. This work is part of the project "Global sea level change since the Mid Holocene: Background trends and climate-ice sheet feedbacks" funded from the Deutsche Forschungsgemeinschaft (DFG) as part of the Special Priority Program (SPP)-1889 "Regional Sea Level Change and Society" (SeaLevel), which funded H. Yang. C. Rodehacke has been supported by the German Federal Ministry of Education and Research (Bundesministerium für Bildung und Forschung, BMBF, grant no. 01LS1612A) and through the National Centre for Climate Research (NCFK, Nationalt Center for Klimaforskning) provided by the Danish State. S. Xu is funded by the China Scholarship Council (CSC).



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
