# Peer review of "The diurnal Energy Balance Model (dEBM): A convenient surface mass balance solution for ice sheets in Earth System modeling"

_The Cryosphere, 2020_

## Referee Comment (RC1) · Anonymous Referee #1 · 2 Dec 2020

This manuscript presents a low-complexity energy and mass balance model for the surface of the Greenland ice sheet. Special attention is given to a parameterization of the diurnal cycle of melting and refreezing, based on a previous implementation of a very similar model, published by the same authors. Different climate boundary conditions are used to test the performance of the model. The text is complete and pleasant to read. The figures are adequate. I think the results are interesting and that this new model is a valuable addition to the existing literature. My comments are mostly minor but they include some questions on the underlying equations. In general, I would like to read more about the expected use cases for this new model. How fast does it run? Is there room for future improvements or would they slow it down too much? If

dEBM is used in the framework of AWI-ESM or other AOGCMs, is the computational speed relevant compared to the climate model? What is the reason for the focus on monthly input data when GCMs can supply daily data? I believe that most of these questions will be very easy to answer but they are valuable as context.

Detailed comments:

figure 2: The distribution appears to also be bimodal in air temperature.

fig. 4, 5: Are good matches in variability and mean mutually exclusive? Why (not)?

fig. 9: dEBM underestimates both melting and refreezing as compared to MAR. What consequences does that have for the sensitivity under different climate boundary conditions?

fig. 10: Albedo too high in dEBM, especially in low accumulation North East where snow aging is most important.

p2, l3ff: You could cite Aschwanden et al. (2019) in this paragraph for showing that calving becomes gradually less important in the future.

p2, l15: The purpose of this sentence is not clear.

p2, l20: Avoid brackets

p2, l26: Please consider referencing van de Berg et al. (2011) and Robinson and Goelzer (2014). van de Berg et al. could also be discussed in section 5.3.

p2, l32: "around zero", replace with "melting point"

p3, l1: The optimal resolution is a matter of debate. I do not think that we can reliably say that 10 km is enough.

p3, l4: Typo: Noël

p3, l 28: "the first section" This should probably read "second" or "following".

p3, l33: remove "used in the following"

p4, l9ff: I found the three paragraphs at the center of the page hard to follow, possibly because they invert the logical progression of the following sections (2.4 -> 2.3 -> 2.2). Also, they contain several assumptions and it is not always clear if they have been tested, maybe in previous work, or if they are otherwise justified.

p4, l15: Within the two clusters there is a clear range of temperatures (Fig. 2), probably related to an implicit spatial signal. Please discuss how this impacts this important assumption. In addition, please discuss how the limited coverage of the PROMICE stations may skew the results.

p4, l31: "implicates" -> "implies" (throughout the paper)

p5, l8: typo: temperature; Does this statement concern monthly average values?

p6, l1: This is a strong statement that requires a reference.

p6, l7: Maybe "upper limit" is a better phrase than "bounded above"?

eq. 4: "mth" is not defined. I suspect is is the month, but that would be inconsistent with equation 14. I suggest to find a suitable symbol. Also, the square brackets are not necessary, I think.

p6, l15: "is" -> "was"

section 2.3: Almost every section starts with a reference to Krebs-Kanzow et al.

eq. 5: Why is the linearization around T0 needed? The presentation of these central equations would greatly benefit from a more detailed description of the individual terms. The term containing "a" contains the downward longwave flux, but I do not understand why this term contains the emissivity of ice. This is also included in Krebs-Kanzow et al. (2018b).

p6, l19: missing: "at THE melting point"

p6, l27: Why not use the LW down forcing directly?

eq. 7: sigma has not yet been defined. This is inconsistent with the Stefan-Boltzmann constant. I am confused over the units of SW, Q, LW, etc. The second line on equation 7 suggests that SW_MP is smaller than SW_fair, because both the fraction of times and the q factor are smaller than one. This makes sense if SW is an energy. However, SW is also used in equation 5, where the second term uses the (linearized) Stefan Boltzmann law. It should thus have the unit power, i.e., energy per unit time.

section 2.4: It might help to move the information about pre-processing further up.

p7, l24: add: "...law DOWNWARD longwave radiation can be..."

p8, l2: I think H_int has not been introduced yet (see line 8).

p8, l16: typo: "."

p8, l19: typo

p9, l4: I think tau_cloudy is missing in the second equation.

eq. 13: So eps_fair and eps_cloudy are both smaller than eps_a?

p10, l9: "PDD" twice

p10, l10: Does that mean that T_MP is not a temperature? The same equality between T and PDD is used in equation 7.

p10, l27: typo

eq. 14: Inconsistent use of dT and "-1". See also "-12" on the second equation on the same page.

p11, l11: remove "choice of the"

p11, l13: typo

p12, l5: This should reference equation 5, not 6.

p12, l6: Does R not have units? Why were no positive values tested?

p12, l8: Please include information about the grid resolution earlier in the ms., e.g., in section 2.4.

p12, l11: typo: "data setS"

p13, l9: units of RMSE missing

p13, l10f: Why are the biases negative. Is this shown in figure 5?

p14, l1: The best value of R is at the extreme of the tested range. See comment above about positive anomalies for R.

p15, l16: typo

p15, l18: The biases cancel out, but their existence indicates that important physical processes are not captured. Please discuss.

p16, l3: "multitude of reasons" I agree, but we need more detail here given the seasonal biases that may point to missing processes that are important under different climate boundary conditions.

p16, l5: I disagree with this assessment. Sub-monthly variations may play a role, but the lack of a snow aging algorithm is likely also important in the dry interior of Greenland. This is consistent with the anomalously high albedo (line 12).

p21, l7: typo

p21, l9: typo

p21, l17: Is this grid different from the one above? Please make this explicit.

p25, l10: typo "surface typeS"

p25, l11: typo

p25, l16: typo

p27, l16: "imply"

References: Aschwanden et al. (2019), Contribution of the Greenland Ice Sheet to sea level over the next millennium, doi:10.1126/sciadv.aav9396

van de Berg et al. (2011), Significant contribution of insolation to Eemian melting of the Greenland ice sheet, doi:10.1038/NGEO1245

Robinson and Goelzer (2014), The importance of insolation changes for paleo ice sheet modeling, doi:10.5194/tc-8-1419-2014

---

## Short Comment (SC1) · 29 Dec 2020

RequirePackagecolor

**Response to Reviewer Comments - Reviewer 1**

We would like to thank Reviewer 1 for the constructive feedback and detailed suggestions. After the open discussion we will provide a point-by-point response. However, the reviewer's comments helped to identify two errors in the manuscript we would

already like to address. In the following the reviewer's comments are printed in blue and our response is printed in black.

eq. 13: So $\epsilon_{fair}$ and $\epsilon_{cloudy}$ are both smaller than $\epsilon_a$?
Thank you for spotting this! There is a typo in the second row of eq. 13 and this should be

$$\epsilon_{cloudy} = \epsilon_a + (1 - CC)\Delta\epsilon.$$

p13, l10f: Why are the biases negative. Is this shown in figure 5?
Unfortunately, Fig. 5 displays the data-model difference but the text refers to the model-data bias. We apologize for this inconsistency. In the revised version Fig. 5 will be corrected accordingly.

---

## Referee Comment (RC2) · Anonymous Referee #2 · 13 Jan 2021

The authors propose a new SMB model to quickly simulate SMB of the Greenland Ice Sheet for a long time (hundreds to millennium). The manuscript is well written, many tables and figures are of good quality. I appreciate the careful preparation of the manuscript. The model performance of dEBM compares favorably with that of the regional climate model. This study will bring new knowledge on the past reconstruction and future projection of SMB of the Greenland Ice Sheet and therefore fall within the scope of The Cryosphere. However, I would like to suggest authors do some modifications before acceptance for publication. Major and specific comments are as below. I hope that my comment is very useful for the improvement of the manuscript.

[Figure]

Major comment:

JJA albedo simulated with dEBM was significant greater in south-western Greenland than that simulated with MAR (Figure 10). This is the reason that dEBM does not consider the effect of dark region (Wientjes et al., 2011) on SMB, which frequently appears on south-western Greenland during summer I guess. Previous studies suggest that the dark region significantly affects the SMB of the GrIS (e.g. Cook et al., 2020). The effect cannot be ignored to evaluate the SMB of the GrIS. dEBM uses the same albedo values (0.55) for ice and wet snow, but it's not realistic to assume an ice albedo of 0.55 in the coastal region. Fig. 13 showed negative SMB simulated with dEBM appeared in the late 21st century, whereas it showed SMB simulated with MAR appeared in the early 21st century. I guess this is due to the overestimation of SMB in the ablation area of the GrIS in the case of dEBM. Because the generation of the dark region is related to microbial activity, the incorporation of the albedo reduction caused by the dark region into dEBM may be still difficult. However, at least, authors should more discuss a factor affecting JJA albedo in Greenland. In addition to that, I suggest authors to more describe future challenges to improve dEBM.

Reference:

Cook, J.M., Tedstone, A. J., Williamson, C., McCutcheon, J., Hodson, A. J., Dayal, A., Skiles, M., Hofer, S., Bryant, R., McAree, O., McGonigle, A., Ryan, J., Anesio, A.M., Irvine-Fynn, T. D. L., Hubbard, A., Hanna, E., Flanner, M., Mayanna, S., Benning, L. G., van As, D., Yallop, M., McQuaid, J. B., Gribbin, T. and Tranter, M. (2020): Glacier algae accelerate melt rates on the southwestern Greenland ice sheet. Cryosphere, 14, 309-330, doi:10.5194/tc-14-309-2020.

Wientjes, I. G. M., Van de Wal, R. S.W., Reichart, G. J., Sluijs, A. and Oerlemans, J. (2011): Dust from the dark region in the western ablation zone of the Greenland ice sheet. Cryosphere, 5, 589-601, doi:10.5194/tc-5-589-2011. 2011.

—

Specific comments:

- 1 Introduction

P. 2 Line 4: Replace "cemtury" with "century".

P. 3 Line 4-5: I suggest adding NHM-SMAP (Niwano et al., 2018), which is a 5km resolution regional climate model, to the list of regional climate models to evaluate SMB of the GrIS.

Reference: Niwano, M., Aoki, T., Hashimoto, A., Matoba, S., Yamaguchi, S., Tanikawa, T., Fujita, K., Tsushima, A., Iizuka, Y., Shimada, R. and Hori, M. (2018): NHM-SMAP: spatially and temporally high-resolution nonhydrostatic atmospheric model coupled with detailed snow process model for Greenland Ice Sheet. The Cryosphere, 12, 635-655.

- 2 Model Description

P. 6 Line 14-15: Please more explain why does this study neglect the effect of sublimation, evaporation, and hoar on SMB of the GrIS. Also, to calculate these properties by dEBM, what atmospheric forcing does dEBM require?

P. 7 Line 5: Why is the albedo differentiated between fair and cloudy sky conditions? In my understanding, albedo is used as a constant value for each surface type in dEBM. Please explain clearly more.

P. 8 Line 5: Could you show me a map of Hice and Hint? Also, how did you get such elevation information? Because spatial interpolation is an important part of this study, the authors should describe the elevation data clearly.

P. 8 Line 13-16: Does dEBM require rainfall and snowfall rate as atmospheric forcings (input data), respectively? In section 2.1, the authors describe that total precipitation rate is used as an atmospheric forcing for dEBM simulation.

P. 8 Line 16: Please replace ". respectively." with ", respectively."

P. 8 Line 20: Is CC in eq. (10) interpolated? If not, please describe the reason not to interpolate CC. If interpolated, please describe the method. LW seems highly dependent on CC according to eq. (10).

P. 9 Line 1-2: How did you classify sky conditions (cloudy and fair) in the other season such as MAM (March, April and May)?

P. 9 Line 7: Isn't "CC > 0.9"?

P. 11 Sub-section 2.7: Can dEBM output the volume of the transformed ice? I think that such spatio-temporal information would be useful to evaluate SMB from the past to the future.

P. 11 Sub-section 2.7: Replace "m" with "mth" because "mth" is used in eq. (4).

P. 11 Sub-section 2.7: Add "(15)" to the later equation.

- 3 Parameter selection and evaluation based on observations

P. 12 Line 6: Replace "(Fettweis et al., 2020)" with "Fettweis et al. (2020)".

P. 12 Line 6-8: It's better to add information on original spatial resolution (before interpolation). P. 12 Line 9-10: Modify italics

P. 13-14 Figures 4 and 5: Could you show me the relationship between the simulated mWE (Gt/yr) and observed mWE (Gt/yr)? I did not understand the messages of Figures 4 and 5 due to much information. Authors should show a model bias for local and GRACE observation, respectively, first.

- 4 Evaluation based on the regional climate model MAR

P. 15 Experimental design: Authors mentioned that dEBM showed good performance in the simulated SMB using atmospheric forcing derived from MAR simulation. In my understanding, dEBM has an advantage of computational time for the SMB simulation compared with MAR. However, Isn't the calculation time of dEBMMAR, ERA more than

that of MARERA? If so, is there any advantage to using dEBMMAR, ERA? MARERA has already shown reasonable performance in SMB in my sense. Please describe this section more carefully and emphasize the advantage of dEBM compared with MAR.

- 5 Sensitivity of the SMB to climate

P. 21 Line 16: Please describe the original spatial resolution of AWI-ESM forcing. Also, how did you get the forcing dataset? Please describe the information on the dataset clearly.

P. 23 Figure. 12: Ice sheet area gradually would change from past (Mid Holocene) to future (2099) I think. Could dEBM simulate the ice sheet area in Greenland? The ice sheet is being retreated under climate warming, so the ice sheet dynamics would significantly affect the SMB of the GrIS. I suggest adding a brief discussion about inter-annual changes in the ice sheet area.

- 6 Conclusion

P. 26 Line 5-7: I'm curious about the computational time of dEBM. Authors should describe the specific time in the manuscript. For example, how long did H6K and Industrial simulation take, respectively?

P. 26 Line 8-9: As I mentioned in the major comment, further study is necessary to accurately evaluate SMB in GrIS, especially the south-western region. Please describe future challenges briefly.

- Appendix A

Table A1: Please add CC as forcing into the table.

- References

P. 33 Line 9-15: The paper has been published on TC. Please replace.

---

## Author Comment (AC1) · 19 Feb 2021

**Response to both reviewers**

We thank both reviewers for their mostly positive and very constructive reviews. Many of their questions raised very interesting points which motivated new analyses and opened new perspectives. We however did not include any new figures in the revised paper as the paper is already quite long, but we are happy to include these in the supplement. We plan to revise our manuscript according to the reviewers' detailed comments (highlighted in blue) as outlined below. We also corrected all typos. Proposed modifications of the manuscript are given in italics.

**1   Response to Reviewer 1**

This manuscript presents a low-complexity energy and mass balance model for the surface of the Greenland ice sheet. Special attention is given to a parameterization of the diurnal cycle of melting and refreezing, based on a previous implementation of a very similar model, published by the same authors. Different climate boundary conditions are used to test the performance of the model. The text is complete and pleasant to read. The figures are adequate. I think the results are interesting and that this new model is a valuable addition to the existing literature. My comments are mostly minor but they include some questions on the underlying equations. In general, I would like to read more about the expected use cases for this new model. How fast does it run? Is there room for future improvements or would they slow it down too much? If dEBM is used in the framework of AWI-ESM or other AOGCMs, is the computational speed relevant compared to the climate model? What is the reason for the focus on monthly input data when GCMs can supply daily data? I believe that most of these questions will be very easy to answer but they are valuable as context.

We have added to the first paragraph of section 6: *"In its Fortran version the computational cost of the actual dEBM is similar to the cost of the necessary interpolations with existing interpolation weights. It takes about 5 seconds to compute the SMB of one year for a configuration with 360000 gridpoints on a Cray CS400. A matlab version of the model simulates the 1979-2016 SMB of the GrIS at 1km resolution in approximately 30 minutes on a Linux desktop PC. Requiring only monthly forcing also provides for an uncomplicated interface, as monthly forcing usually is more accessible in case of completed transient climate simulations such as simulations of the CMIP5 project (Taylor et al., 2012)."* We also list some possible use cases at the end of the same section.

**1.1   Response to Detailed Comments**

figure 2: The distribution appears to also be bimodal in air temperature.

In the PROMICE data, the temperature distribution does not appear to be particularly bimodal. We have analysed PROMICE weather station data and defined monthly mean air temperatures $T_{fair}$ and $T_{cloudy}$ analogously to $\tau_{fair}$ and $\tau_{cloudy}$, and we do not find a clear separation between the two modes (Fig. AC1-1).

fig. 4, 5: Are good matches in variability and mean mutually exclusive? Why (not)?

They are, and we propose that this is due to some systematic bias in the precipitation forcing. Due to the relative coarse resolution of the ERA-Interim reanalysis, it is plausible that the interpolated precipitation forcing systematically underestimates small-scaled maxima in accumulation over the margins of the ice sheet. In consequence, optimizing our scheme to the mean SMB would likely provide optimal parameters, that underestimate melting in compensation. On the other hand, the SMB variability and the multi-year trend will be primarily influenced by melt variability and will be less sensitive to such systematic bias related to the resolution of the forcing. Thus, we here decided to optimize the scheme only to spatial and temporal variations around the respective means of the local and integral observations. This rationale is supported by the finding that using MAR precipitation of higher resolution results in a substantially lower bias to the mean local and integral observations.

We have improved the respective paragraph at the end of section 3.1, which was intended to motivate this aspect of our calibration strategy and moved it to the beginning of section 3.2: *"For the evaluation of the dEBM scheme we take into account that the precipitation forcing is possibly biased: as precipitation is interpolated from the coarse resolution ERA-Interim data, the intensified snow accumulation at the slopes and margins of the ice sheet may be systematically underestimated. Also low accumulation rates in the interior may be relatively inaccurate in the ERA-Interim reanalysis. "*

fig. 9: dEBM underestimates both melting and refreezing as compared to MAR. What consequences does that have for the

[Figure]

**Figure AC2-1.** Monthly mean air temperatures of cloudy and fair days, $T_{cloudy}, T_{fair}$ as a function of the respective transmissivity $\tau_{cloudy}, \tau_{fair}$ of all summer months as calculated from up to 11 years of daily observations from 17 PROMICE weather station. Every symbol represents the respective parameters as diagnosed for one individual month at one station. Colors reflect the respective monthly mean emissivities of cloudy and fair days respectively.

sensitivity under different climate boundary conditions?
Due to its monthly forcing and crude representation of the snow pack, dEBM has obvious limitations near the transition from ablation to accumulation zones. On continental-scale ice sheets we expect that these limitations are mostly restricted to the upper boundary of the ablation zone, being narrower where the slope is steep. As the upper ablation zone tends to become steeper
5  in global warming scenarios (Goelzer et al., 2020) we expect that uncertainties in the transition zone will not increase with warmer climates. To some extent these considerations are already part of the discussion, and given the length of the manuscript we would like to not extend it any further.
fig. 10: Albedo too high in dEBM, especially in low accumulation North East where snow aging is important.
This is true and we address this in the last paragraph of section 4.2. We have take this up and now state: *"...while albedo is*
10  *mostly overestimated in the North where accumulation rates are low and snow aging is most important."*
p2, l3ff: You could cite Aschwanden et al. (2019) in this paragraph for showing that calving becomes gradually less important in the future.
Will do.
p2, l15: The purpose of this sentence is not clear
15  We rephrased the preceding and following sentence: *"However, changes in insolation due to long-term changes in the Earth's orbit can influence the sensitivity of the SMB to temperature (van de Berg et al., 2011; Robinson and Goelzer, 2014). Also, field measurements from glaciers outside of Greenland reveal that optimal parameters for the PDD scheme strongly differ for different latitudes, altitudes, or climate zones (Hock, 2003). Therefore it is questionable if the empirical, Greenland based parameterization can be applied to ice sheets outside of Greenland (e.g. the ice sheets of the last ice age) or in different climates."*
20  p2, l20: Avoid brackets
OK.

p2, l26: Please consider referencing van de Berg et al. (2011) and Robinson and Goelzer (2014). van de Berg et al. could also be discussed in section 5.3.

We do so now.

p2, l32: "around zero", replace with "melting point"

OK

p3, l1: The optimal resolution is a matter of debate. I do not think that we can reliably say that 10 km is enough.

We change this to *"Therefore, melting occurs in a narrow strip along the ice sheets' margins which requires a resolution that is still beyond the scope of multidecadal global climate simulations or reanalysis products."*

p3, l4: Typo: Noël

OK.

p3, l 28: "the first section" This should probably read "second" or "following"

We have changed this to the "following".

p3, l33: remove "used in the following"

OK.

p4, l9ff: I found the three paragraphs at the center of the page hard to follow, possibly because they invert the logical progression of the following sections (2.4 -> 2.3 -> 2.2). Also, they contain several assumptions and it is not always clear if they have been tested, maybe in previous work, or if they are otherwise justified.

We tried to improve and shorten this paragraph, however, we kept the approach to first outline the general concept in a bottom-up fashion, to then develop the model top-down.

p4, l15: Within the two clusters there is a clear range of temperatures (Fig. 2), probably related to an implicit spatial signal. Please discuss how this impacts this important assumption. In addition, please discuss how the limited coverage of the PROMICE stations may skew the results.

The range of temperatures is predominantly related to the elevation range of the PROMICE stations (Fig. AC1-2). The cloud thickness may be reduced at high elevations and $\tau_{cloudy}$ is therefore elevation dependent. For $\tau_{fair}$ (the empirical parameter in our downscaling) the elevation effect is small by comparison. The temperature dependence in emissivities $\epsilon_{fair,cloudy}$ is in part related to the larger water vapor content of warmer air and is implicitly accounted for, as we do not constrain $\epsilon_{fair}$ but only prescribe $\Delta\epsilon$. Note, that the fractionated radiative fluxes of the cloudy and fair days are always consistent with the monthly mean forcing and the scheme is actually not very sensitive to the specific choice of its empirical parameter $\Delta\epsilon$ and $\tau_{fair}$ as long as these are modified consistently (e.g. if we assume that fair days are completely cloud free with $\tau_{fair} \approx 0.8$, then cloudy days must be also completely overcast and $\Delta\epsilon \approx 0.21$, Sedlar and Hock (2009)).

p4, l31: "implicates" -> "implies" (throughout the paper)

OK

p5, l8: typo: temperature; Does this statement concern monthly average values?

OK, and yes we now specify that this is monthly mean temperature, here.

p6, l1: This is a strong statement that requires a reference.

We changed this to: *as outgoing longwave radiation usually dominates the energy balance of a cold surface in clear nights (section 2.3).*

p6, l7: Maybe "upper limit" is a better phrase than "bounded above"

We changed this to "limited"

eq. 4: "mth" is not defined. I suspect is is the month, but that would be inconsistent with equation 14. I suggest to find a suitable symbol. Also, the square brackets are not necessary, I think.

We now use m for month throughout the text and also added m to the appendix.

p6, l15: "is" -> "was"

This sentence was replaced by *"Other contributions to the SMB such as sublimation, evaporation, and hoar are so far neglected by the dEBM as it is not expected that our downscaling approach can improve the respective mass fluxes from climate models, which simulate these processes on larger spatial scales but shorter time steps. With minor technical modifications, these fluxes can be individually added to snow fall $SF$ and rain fall $RF$ as an additional forcing (negative net snow fall will not pose a problem)."*

[Figure]

**Figure AC2-2.** Monthly mean emissivity of cloudy and fair days, $\epsilon_{cloudy}, \epsilon_{fair}$ as a function of the respective transmissivity $\tau_{cloudy}, \tau_{fair}$ of all summer months as calculated from up to 11 years of daily observations from 17 PROMICE weather station. Every symbol represents the respective parameters as diagnosed for one individual month at one station. Colors reflect the elevation of the respective weather station.

section 2.3: Almost every section starts with a reference to Krebs-Kanzow et al
We have removed two.
eq. 5: Why is the linearization around T0 needed? The presentation of these central equations would greatly benefit from a more detailed description of the individual terms. The term containing "a" contains the downward longwave flux, but I do not
5   understand why this term contains the emissivity of ice. This is also included in Krebs-Kanzow et al. (2018b).
The linearization is not needed indeed, but does not substantially affect the accuracy and simplifies the preprocessing and estimation of the elevation angle. We kept it because it allows to directly compare our scheme to empiric temperature index schemes. As the melt scheme is already published, we do not want to go into detail here. The emissivity of ice comes into play as we assume that the ice absorbs longwave radiation according to its emissivity (a good absorber is also a good emitter).
10  p6, l19: missing: "at THE melting point"
OK.
p6, l27: Why not use the LW down forcing directly?
The emissivity needs to be downscaled via emissivities to be consistent with downscaled temperature, also emissivities are needed to fractionate the radiative forcing.
15  eq. 7: sigma has not yet been defined. This is inconsistent with the Stefan-Boltzmann constant.
We have changed the nomenclature and now define $PDD_{3.5}$ in section 2.4.
eq. 7: I am confused over the units of SW, Q, LW, etc. The second line on equation 7 suggests that $SW_{MP}$ is smaller than $SW_{fair}$, because both the fraction of times and the q factor are smaller than one. This makes sense if SW is an energy. However, SW is also used in equation 5, where the second term uses the (linearized) Stefan Boltzmann law. It should thus have the
20  unit power, i.e., energy per unit time.
This was indeed not clear, we now state explicitly that *"$Q_{MP}$ represents a monthly mean energy flux"*
and rephrased the following paragraph to*"$T_{MP}$ is the near-surface temperature $T_{MP}$ during the melt period. $T_{MP}$ is param-*

*eterized by the positive-degree days per month as defined in section 2.4. The $\Delta t_\Phi$ is the length of the melt period when the sun exceeds the elevation angle $\Phi$. The ratio $\frac{\Delta t_\Phi}{\Delta t}$ converts the energy flux during the melt period to daily fluxes. The $q_\Phi$ is the ratio $\frac{SW_\Phi}{SW_0}$ (surface short-wave radiation averaged over the daily melt period relative to short-wave radiation averaged over the whole day). The parameters $q_\Phi$ and $\Delta t_\Phi$ are functions of the elevation angle $\Phi$, which is here calculated locally as we use spatially variable atmospheric emissivity."*

section 2.4: It might help to move the information about pre-processing further up.

We would like to keep the top-down structure here.

p7, l24: add: "...law DOWNWARD longwave radiation can be..."

OK

p8, l2: I think $H_int$ has not been introduced yet (see line 8).

We now introduce $H_{cr}$, $H_{int}$ and $H$ together with the forcing in section 2.1.

p9, l4: I think $\tau_{cloudy}$ is missing in the second equation.

No, we don't need $\tau_{cloudy}$ here, as we can diagnose $SW_{cloudy}$ from $SW_{fair}$, $SW$ and $CC$ according to the fractionation. We understand that this paragraph was hard to understand and we have reformulated the first part:

*"We fractionate downward longwave radiation and shortwave radiation for fair and cloudy conditions:*

$$
\begin{aligned}
SW\downarrow &= CC\,SW^{\downarrow}_{cloudy} + (1-CC)SW^{\downarrow}_{fair}\\
LW\downarrow &= CC\,LW^{\downarrow}_{cloudy} + (1-CC)LW^{\downarrow}_{fair}
\end{aligned}
\tag{1}
$$

*To avoid numeric problems, we only apply this separation, if monthly cloud cover is in the range of [0.1 0.9] and otherwise use unseparated $SW\downarrow$ and $LW\downarrow$ to calculate the energy balance $Q$, accounting (not accounting) for the diurnal melt period during the entire month, if $CC < 0.1$ ($CC > 0.9$), respectively.*

*Where we separate cloudy and fair conditions, we need to introduce two additional assumptions which here are based on an analysis of PROMICE automatic weather station data (Ahlstrom et al., 2008). Specifically we analyze daily radiation, cloud cover, and air temperature observations from 17 stations, which cover up to 11 years (Fig. 2). Applying equation 8 we diagnose distinct atmospheric emissivities $\epsilon_{fair}$ and $\epsilon_{cloudy}$ for fair or cloudy conditions and similarly we diagnose atmospheric transmissivities $\tau_{fair}$ and $\tau_{cloudy}$ according to*

$$
SW^{\downarrow}_{fair,cloudy} = \tau_{fair,cloudy}\widehat{SW}
\tag{2}
$$

*with atmospheric transmissivities $\tau_{cloudy}$ and $\tau_{fair}$ for fair and cloudy conditions.*

*...*

*To separate longwave radiation, we constrain atmospheric emissivities by defining $\Delta\epsilon$ to be the emissitvity increase due to cloud cover with*

$$
\epsilon_{cloudy} = \epsilon_{fair} + \Delta\epsilon.
\tag{3}
$$

*."* eq. 13: So $\epsilon_{fair}$ and $\epsilon_{cloudy}$ are both smaller than $\epsilon_a$?

Thank you for spotting this! Of cause there is a typo in the second equation and this should be:

$$
*\quad
\begin{aligned}
\epsilon_{fair} &= \epsilon_a - CC\,\Delta\epsilon\\
\epsilon_{cloudy} &= \epsilon_a + (1-CC)\Delta\epsilon.
\end{aligned}
\tag{4}
$$

p10, l9: "PDD" twice

We rephrased this.

p10, l10: Does that mean that $T_M P$ is not a temperature? The same equality between T and PDD is used in equation 7.

We now clearly define PDD to be the positive-degree days per month.

eq. 14: Inconsistent use of dT and "-1". See also "-12" on the second equation on the same page

We now use $\Delta t_m$ as the length of month m throughout of the paper.

p11, l11: remove "choice of the"

We replaced it with *"The albedo parameterisation..."*.

p12, l5: This should reference equation 5, not 6.

corrected.

p12, l6: Does R not have units? Why were no positive values tested?

We added W/m2. We introduced R to primarily account for the heat flux to lower layers, that is why it was initially tested for negative values only. We have now extended the range of R to $[-2, -1, 0, 1, 2] \mathrm{W\,m^{-2}}$ and updated the figures accordingly.

p12, l8: Please include information about the grid resolution earlier in the ms., e.g., in section 2.4.

We now include the necessity to define a target grid in section 2.1: *"Furthermore a target grid of sufficient resolution needs to be defined and respective high-resolution surface elevation data H need to be available."*
The model does not require a specific resolution.

p13, l10f: Why are the biases negative. Is this shown in figure 5?

Unfortunately, Fig. 5 displays the data-model difference but the text refers to the model-data bias. We apologize for this inconsistency. In the revised version Fig. 5 will be corrected accordingly.

p14, l1: The best value of R is at the extreme of the tested range. See comment above about positive anomalies for R.

See reply above.

p15, l18: The biases cancel out, but their existence indicates that important physical processes are not captured. Please discuss.

We added *" and missing processes such as snow aging, which may particularly bias late summer melt"*

p16, l3: "multitude of reasons" I agree, but we need more detail here given the seasonal biases that may point to missing processes that are important under different climate boundary conditions.

We provide a list of processes which we expect to be most relevant *"the simplicity of the dEBM albedo scheme, unresolved sub-monthly variability or the (neglected) effect of humidity and high wind speed on turbulent heat fluxes which will be important at coastal locations."* and also discuss missing processes in section 6. Furthermore, it is somewhat promising that the extreme year of 2012 is well represented in our simulation (Fig. 6). Even though the model demonstrates a good skill, one should keep in mind that it was originally designed to replace the most simplistic PDD interface, which is still frequently used in ice sheet modelling on multi-millennial timescales. We therefore now state at the beginning of section 2.1: *"The model is formulated with a focus on the ablation zone; if surface conditions do not favour surface melt, the surface mass balance is exclusively controlled by the accumulation of snow."*

p16, l5: I disagree with this assessment. Sub-monthly variations may play a role, but the lack of a snow aging algorithm is likely also important in the dry interior of Greenland. This is consistent with the anomalously high albedo (line 12).

True, snow aging will be important in the interior and we have extended this sentence to: *" This is likely related to unrepresented sub-monthly temperature variability, as temperatures exhibit stronger variability at high elevations (Fausto et al., 2011) and the constant albedo for dry snow which cannot account for snow aging in low-accumulation regions of the interior GrIS."*
It should be however noted that the albedo scheme includes some kind of memory effect. We therefore added to section 2.6: *"The scheme first tests whether the new snow of that month is likely to survive. If this is not the case the scheme includes some element of persistence: if snow was wet (dry or new) in the previous month it is first tested whether conditions allow that the surface remains wet (dry).*

p21, l17: Is this grid different from the one above? Please make this explicit.

Yes, we changed this to: *"The AWI-ESM forcing is here downscaled to an equidistant 5km grid in contrast to the 1km grid used in the previous section."*

**References**

Ahlstrom, A. P., Gravesen, P., Andersen, S. B., van As, D., Citterio, M., Fausto, R. S., Nielsen, S., Jepsen, H. F., Kristensen, S. S., Christensen, E. L., Stenseng, L., Forsberg, R., Hanson, S., Petersen, D., and Team, P. P.: A new programme for monitoring the mass loss of the Greenland ice sheet, Geological Survey of Denmark and Greenland Bulletin, pp. 61–64, 2008.

5  Fausto, R. S., Ahlstrom, A. P., Van As, D., and Steffen, K.: Present-day temperature standard deviation parameterization for Greenland, Journal of Glaciology, 57, 1181–1183, 2011.

Goelzer, H., Nowicki, S., Payne, A., Larour, E., Seroussi, H., Lipscomb, W. H., Gregory, J., Abe-Ouchi, A., Shepherd, A., Simon, E., Agosta, C., Alexander, P., Aschwanden, A., Barthel, A., Calov, R., Chambers, C., Choi, Y., Cuzzone, J., Dumas, C., Edwards, T., Felikson, D., Fettweis, X., Golledge, N. R., Greve, R., Humbert, A., Huybrechts, P., Le clec'h, S., Lee, V., Leguy, G., Little, C., Lowry, D. P., Morlighem,

10  M., Nias, I., Quiquet, A., Rückamp, M., Schlegel, N.-J., Slater, D., Smith, R., Straneo, F., Tarasov, L., van de Wal, R., and van den Broeke, M.: The future sea-level contribution of the Greenland ice sheet: a multi-model ensemble study of ISMIP6, The Cryosphere Discussions, 2020, 1–43, https://doi.org/10.5194/tc-2019-319, https://tc.copernicus.org/preprints/tc-2019-319/, 2020.

Hock, R.: Temperature index melt modelling in mountain areas, Journal of Hydrology, 282, 104–115, https://doi.org/10.1016/S0022-1694(03)00257-9, 2003.

15  Robinson, A. and Goelzer, H.: The importance of insolation changes for paleo ice sheet modeling, The Cryosphere, 8, 1419–1428, https://doi.org/10.5194/tc-8-1419-2014, https://tc.copernicus.org/articles/8/1419/2014/, 2014.

Sedlar, J. and Hock, R.: Testing longwave radiation parameterizations under clear and overcast skies at Storglaciären, Sweden, The Cryosphere, 3, 75–84, https://doi.org/10.5194/tc-3-75-2009, https://www.the-cryosphere.net/3/75/2009/, 2009.

Taylor, K. E., Stouffer, R. J., and Meehl, G. A.: An overview of CMIP5 and the experiment design, Bulletin of the American Meteorological

20  Society, 93, 485–498, 2012.

van de Berg, W. J., van den Broecke, M., Ettema, J., van Meijgaard, E., and Kaspar, F.: Significant contribution of insolation to Eemian melting of the Greenland ice sheet, Nature Geoscience, 4, 679–683, https://doi.org/10.1038/NGEO1245, 2011.

---

## Author Comment (AC2) · 19 Feb 2021

**Response to both reviewers**

We thank both reviewers for their mostly positive and very constructive reviews. Many of their questions raised very interesting points which motivated new analyses and opened new perspectives. We however did not include any new figures in the revised paper as the paper is already quite long, but we are happy to include these in the supplement. We plan to revise our manuscript according to the reviewers' detailed comments as outlined below. We also corrected all typos. Proposed modifications of the manuscript are given in italics.

**1 Response to Reviewer 2**

The authors propose a new SMB model to quickly simulate SMB of the Greenland Ice Sheet for a long time (hundreds to millennium). The manuscript is well written, many tables and figures are of good quality. I appreciate the careful preparation of the manuscript. The model performance of dEBM compares favorably with that of the regional climate model. This study will bring new knowledge on the past reconstruction and future projection of SMB of the Greenland Ice Sheet and therefore fall within the scope of The Cryosphere. However, I would like to suggest authors do some modifica- tions before acceptance for publication. Major and specific comments are as below. I hope that my comment is very useful for the improvement of the manuscript.

Major comment:

JJA albedo simulated with dEBM was significant greater in south-western Greenland than that simulated with MAR (Figure 10). This is the reason that dEBM does not consider the effect of dark region (Wientjes et al., 2011) on SMB, which frequently appears on south-western Greenland during summer I guess. Previous studies suggest that the dark region significantly affects the SMB of the GrIS (e.g. Cook et al., 2020). The effect cannot be ignored to evaluate the SMB of the GrIS. dEBM uses the same albedo values (0.55) for ice and wet snow, but it's not realistic to assume an ice albedo of 0.55 in the coastal region. Fig. 13 showed negative SMB simulated with dEBM appeared in the late 21st century, whereas it showed SMB simulated with MAR appeared in the early 21st century. I guess this is due to the overestimation of SMB in the ablation area of the GrIS in the case of dEBM. Because the generation of the dark region is related to microbial activity, the incorporation of the albedo reduction caused by the dark region into dEBM may be still difficult. However, at least, authors should more discuss a factor affecting JJA albedo in Greenland. In addition to that, I suggest authors to more describe future challenges to improve dEBM. You are actually raising two very important points here: (i) indeed I don't see how we could explicitly include processes such as microbial activity or dust deposition on the ice sheet in such a simple model. However, also with respect to the important role of dust in the course of the termination of the last ice age, we actually currently consider to prescribe a background bare ice albedo wherever the multi-year SMB is negative and the monthly snow height vanishes. In case of present-day Greenland one could then also prescribe observed melt season albedo in the ablation zone. We added this thought to section 6: *Furthermore one might prescribe a background bare ice albedo to account for regional darkening due to dust deposition or microbial activity (Wientjes et al., 2011; Cook et al., 2020).*

(ii) The fact that we do not see a negative SMB in the first half of this century points to a central problem in global climate modeling which cannot be attributed to or solved by a surface mass balance model. We address this at the end of section 5.2: *"The climate model however does not reproduce the extreme Greenland blocking in the 2005–2015 period, which is a common problem in global climate models (Hanna et al., 2018). Accordingly the interannual variations in SMB of recent decades is underestimated and the simulated negative trend in SMB may be delayed."*

Specific comments:
- 1 Introduction
P. 2 Line 4: Replace "cemtury" with "century".
OK
P. 3 Line 4-5: I suggest adding NHM-SMAP (Niwano et al., 2018), which is a 5km resolution regional climate model, to the list of regional climate models to evaluate SMB of the GrIS.

OK

- 2 Model Description

P. 6 Line 14-15: Please more explain why does this study neglect the effect of sublimation, evaporation, and hoar on SMB of the GrIS. Also, to calculate these properties by dEBM, what atmospheric forcing does dEBM require?

We have here added the following sentences: *"Other contributions to the SMB such as sublimation, evaporation, and hoar are so far neglected by the dEBM as it is not expected that our downscaling approach can improve the respective mass fluxes from climate models, which simulate these processes on larger spatial scales but shorter time steps. With minor technical modifications, these fluxes can be individually added to snow fall $SF$ and rain fall $RF$ as an additional forcing (negative snow fall does not pose a problem)."* We will test this strategy in upcoming studies.

P. 7 Line 5: Why is the albedo differentiated between fair and cloudy sky conditions? In my understanding, albedo is used as a constant value for each surface type in dEBM. Please explain clearly more.

This was not very clear and inconsistent at several places. We have made several modifications: (i) we introduce albedo in 2.3 as *"albedo ($A(SurfaceType)$ which is chosen according to the given surface types (i.e., NewSnow, DrySnow, or WetSnow) and further differentiate these for cloudy and fair conditions following Willeit and Ganopolski (2018)"* (ii) To section 2.6 we have added *"Each surface type is assigned a pair of albedo values for fair and cloudy conditions. Following Willeit and Ganopolski (2018) we assume that the albedo for cloudy conditions exceeds by 0.05 the respective albedo for fair conditions of the same surface type."* and (iii) in section 3.1 it is now *"For fair conditions we vary $A_{NewSnow}$ within $[0.84, 0.845, 0.85]$, $A_{DrySnow}$ within $[0.68, 0.69, 0.70, ..., 0.78]$, $A_{WetSnow}$ within $[0.53, 0.54, 0.55, 0.56, 0.57]$, the albedo values for cloudy conditions are varied with accordingly larger base values, and R varies within $[-2, -1, 0, 1, 2]\mathrm{W\,m^{-2}}$."*

P. 8 Line 5: Could you show me a map of Hice and Hint? Also, how did you get such elevation information? Because spatial interpolation is an important part of this study, the authors should describe the elevation data clearly.

We now specify the resolution of the atmospheric forcing in the manuscript in section 5 *"Both simulations have been conducted with the AWI Earth System Model, AWI-ESM (Sidorenko et al., 2015) at an atmospheric resolution of approximately 1.85X1.85 degree horizontal resolution with 47 vertical levels (T63L47) and both experiments are using an invariant present day ice sheet geometry as boundary conditions."*, also please see Fig. AC2-1

P. 8 Line 16: P. 8 Line 16: Please replace ". respectively." with ", respectively"

OK

P. 8 Line 20: Is CC in eq. (10) interpolated? If not, please describe the reason not to interpolate CC. If interpolated, please describe the method. LW seems highly dependent on CC according to eq. (10).

CC is interpolated (section 2.1 first paragraph)

P. 9 Line 1-2: How did you classify sky conditions (cloudy and fair) in the other season such as MAM (March, April and May)?

We did not analyse the other seasons here because we wanted to find parameters which represent the main melt season.

P. 9 Line 7: Isn't "CC > 0.9"?

Thank you, we corrected this.

P. 11 Sub-section 2.7: Can dEBM output the volume of the transformed ice? I think that such spatio-temporal information would be useful to evaluate SMB from the past to the future.

The model does not simulate snow or ice density. The output of the model would only allow to diagnose the mass of the transformed ice.

P. 11 Sub-section 2.7: Replace "m" with "mth" because "mth" is used in eq. (4).

We replaced mth with m instead.

P. 11 Sub-section 2.7: Add "(15)" to the later equation.

OK.

- 3 Parameter selection and evaluation based on observations

P. 12 Line 6: Replace "(Fettweis et al., 2020)" with "Fettweis et al. (2020)".

OK.

P. 12 Line 6-8: It's better to add information on original spatial resolution (before interpolation).

We did so.

P. 12 Line 9-10: Modify italics

[Figure]

**Figure AC2-1.** Orographies as used in the manuscript: ERA-Interim linearly interpolated to the 1km ISMIP6 coordinates (upper left), ISMIP6 topography (Nowicki et al., 2016) (upper right), AWI-ESM linearly interpolated to an equidistant 5km grid (lower left) and ISMIP6 topography linearly interpolated to the same 5km grid as used for AWI-ESM.

[Figure]

**Figure AC2-2.** Local SMB as simulated by experiment $dEBM_{MAR,ERA}$ as a function of the SMB observations.

OK.

P. 13-14 Figures 4 and 5: Could you show me the relationship between the simulated mWE (Gt/yr) and observed mWE (Gt/yr)? I did not understand the messages of Figures 4 and 5 due to much information. Authors should show a model bias for local and GRACE observation, respectively, first.

5 Figures 4 and 5 illustrate statistics of all calibration experiments (830 experiments). The purpose of these figures is to demonstrate that (i) the systematic bias in the two observational data sets appears to be small (p.13, l.5), (ii) good matches in variability and mean are mutually exclusive and we provide some justification why we chose parameters which provide a good match with the variability. (iii) that using higher resolution precipitation substantially improves the match to the mean and to the variability of the local observations which supports our hypothesis that the precipitation forcing is systematically biased.

10 Fig. AC2-2 illustrates the relationship between simulated and observed local SMB measurement for the experiment $dEBM_{MAR,ERA}$ which was used in section 4.2., Fig. 6 in the manuscript shows the comparison between experiment $dEBM_{MAR,ERA}$ (red), integral observations (black) and MAR(blue).

15

P. 15 Experimental design: Authors mentioned that dEBM showed good performance in the simulated SMB using atmo-
spheric forcing derived from MAR simulation. In my understanding, dEBM has an advantage of computational time for the SMB simulation compared with MAR. However, Isn't the calculation time of dEBMMAR, ERA more than that of MARERA? If so, is there any advantage to using dEBMMAR, ERA? MARERA has already shown reasonable performance in SMB in my sense. Please describe this section more carefully and emphasize the advantage of dEBM compared with MAR.

The sole purpose here was to evaluate the model independent of the suspected bias in precipitation forcing. Regional models
20 can be expected to be superior to dEBM in all respects (apart from their computational cost) and experiment $dEBM_{MAR,ERA}$ does not represent a typical use case; typically one would use coarse resolution climate model output as in section 5.

P. 21 Line 16: Please describe the original spatial resolution of AWI-ESM forcing. Also, how did you get the forcing dataset?

Please describe the information on the dataset clearly.

We now specify that the resolution of the atmospheric component has a *"horizontal resolution of approximately 1.85X1.85 degree with 47 vertical levels (T63L47)."* With respect to their forcing, the global climate simulations for Mid-Holocene and Preindustrial only differ in orbital parameters and greenhouse gas concentrations, following PMIP protocols (Otto-Bliesner et al., 2017) and CMIP5 protocols (Taylor et al., 2012) respectively.

P. 23 Figure. 12: Ice sheet area gradually would change from past (Mid Holocene) to future (2099) I think. Could dEBM simulate the ice sheet area in Greenland? The ice sheet is being retreated under climate warming, so the ice sheet dynamics would significantly affect the SMB of the GrIS. I suggest adding a brief discussion about inter- annual changes in the ice sheet area.

The simulation of changes in the ice sheet area would require to couple dEBM to an ice sheet model. This is of cause an important next step but not the focus of this study. We indicate this in the last part of section 6: *"dEBM can be ... coupled to an ice sheet model using forcing derived from climate models and observation as in Niu et al. (2019)."*

P. 26 Line 5-7: I'm curious about the computational time of dEBM. Authors should describe the specific time in the manuscript. For example, how long did H6K and Industrial simulation take, respectively?

We have added the following lines: *"In its Fortran version the computational cost of the actual dEBM is similar to the cost of the necessary interpolations with existing interpolation weights. It takes about 5 seconds to compute the SMB of one year for a configuration with 360000 gridpoints on a CRAY CS400. A matlab version of the model simulates the 1979-2016 SMB of the GrIS at 1km resolution (approximately 4.8 million grid points) in approximately 30 minutes on a Linux desktop PC. Requiring only monthly forcing also provides for an uncomplicated interface, as monthly forcing usually is more accessible in case of completed transient climate simulations such as simulations of the CMIP5 project (Taylor et al., 2012)."*

P. 26 Line 8-9: As I mentioned in the major comment, further study is necessary to accurately evaluate SMB in GrIS, especially the south-western region. Please describe future challenges briefly.

We added : *"Furthermore one might prescribe a background bare ice albedo to account for regional darkening due to dust deposition or microbial activity (Wientjes et al., 2011; Cook et al., 2020)."*

Table A1: Please add CC as forcing into the table.

OK.

P. 33 Line 9-15: The paper has been published on TC. Please replace.

OK.

**References**

Cook, J. M., Tedstone, A. J., Williamson, C., McCutcheon, J., Hodson, A. J., Dayal, A., Skiles, M., Hofer, S., Bryant, R., McAree, O., McGonigle, A., Ryan, J., Anesio, A. M., Irvine-Fynn, T. D. L., Hubbard, A., Hanna, E., Flanner, M., Mayanna, S., Benning, L. G., van As, D., Yallop, M., McQuaid, J. B., Gribbin, T., and Tranter, M.: Glacier algae accelerate melt rates on the south-western Greenland Ice Sheet, The Cryosphere, 14, 309–330, https://doi.org/10.5194/tc-14-309-2020, https://tc.copernicus.org/articles/14/309/2020/, 2020.

Hanna, E., Fettweis, X., and Hall, R. J.: Brief communication: Recent changes in summer Greenland blocking captured by none of the CMIP5 models, Cryosphere, 12, 3287–3292, https://doi.org/10.5194/tc-12-3287-2018, 2018.

Niu, L., Lohmann, G., Hinck, S., Gowan, E. J., and Krebs-Kanzow, U.: The sensitivity of Northern Hemisphere ice sheets to atmospheric forcing during the last glacial cycle using PMIP3 models, JOURNAL OF GLACIOLOGY, 65, 645–661, https://doi.org/10.1017/jog.2019.42, 2019.

Nowicki, S. M. J., Payne, A., Larour, E., Seroussi, H., Goelzer, H., Lipscomb, W., Gregory, J., Abe-Ouchi, A., and Shepherd, A.: Ice Sheet Model Intercomparison Project (ISMIP6) contribution to CMIP6, Geoscientific Model Development, 9, 4521–4545, https://doi.org/10.5194/gmd-9-4521-2016, 2016.

Otto-Bliesner, B. L., Braconnot, P., Harrison, S. P., Lunt, D. J., Abe-Ouchi, A., Albani, S., Bartlein, P. J., Capron, E., Carlson, A. E., Dutton, A., et al.: The PMIP4 contribution to CMIP6–Part 2: Two interglacials, scientific objective and experimental design for Holocene and Last Interglacial simulations, Geoscientific Model Development, 10, 3979–4003, 2017.

Sidorenko, D., Rackow, T., Jung, T., Semmler, T., Barbi, D., Danilov, S., Dethloff, K., Dorn, W., Fieg, K., Gößling, H. F., et al.: Towards multi-resolution global climate modeling with ECHAM6–FESOM. Part I: model formulation and mean climate, Climate Dynamics, 44, 757–780, 2015.

Taylor, K. E., Stouffer, R. J., and Meehl, G. A.: An overview of CMIP5 and the experiment design, Bulletin of the American Meteorological Society, 93, 485–498, 2012.

Wientjes, I. G. M., Van de Wal, R. S. W., Reichart, G. J., Sluijs, A., and Oerlemans, J.: Dust from the dark region in the western ablation zone of the Greenland ice sheet, The Cryosphere, 5, 589–601, https://doi.org/10.5194/tc-5-589-2011, https://tc.copernicus.org/articles/5/589/2011/, 2011.

Willeit, M. and Ganopolski, A.: The importance of snow albedo for ice sheet evolution over the last glacial cycle, Climate of the Past, 14, 697–707, https://doi.org/10.5194/cp-14-697-2018, 2018.

---

## Author Response (AR2)

**Response to the editor and the reviewers**

We would like to thank both the editor and the reviewers for their comments. We have revised our manuscript according to the reviewers' detailed comments (highlighted in blue) as outlined below. We also modified the acknowledgements. Proposed modifications of the manuscript are given in italics.

5 **Detailed response**

Reviewer 1, 1) The information on computational speed is incomplete. Does the speed on the Cray CS400 refer to using the entire cluster or a single node or single CPU? Rather than naming a specific manufacturer, a statement on the number of CPUs would be sufficient and more useful. The specific model of processor does not matter either, because the reader can assume that it is fairly recent.

10 We now specify the computational cost as follows: *In its Fortran version the actual dEBM code runs as sequential code on one core. After interpolation to the target grid, it takes about 5 seconds to compute the SMB of one year for a configuration with 360000 gridpoints on a CPU core (Xeon Broadwell CPU; E5-2697v4, 2.3 GHz).*

Reviewer 1, 2) reply to my original comment on p4, l15 and figure 2: Please consider adding this information to the manuscript.

We added the following to p. 10 ll. 8-12: *In Fig. 2 parameters reveal a temperature dependence which is predominantly asso-*

15 *ciated with the elevation range of the PROMICE stations. The cloud thickness may be reduced at high elevations and $\tau_{cloudy}$ is therefore elevation dependent. For $\tau_{fair}$ (the empirical parameter in our downscaling) the elevation effect is small by comparison. The temperature dependence in emissivities $\epsilon_{fair,cloudy}$ is in part related to the larger water vapor content of warmer air and is implicitely accounted for, as we do not constrain $\epsilon_{fair}$ but only prescribe $\Delta\epsilon$.*

Reviewer 1, 3) my original comment on equation 13: The typo still is in the manuscript (Version 4).

20 We corrected this (this is unfortunately not highlighted in the tracked-changes file).

Editor: In addition to the points raised by the anonymous referee 1, I would like to suggest that the title of Sect. 6 "Conclusion" is not appropriate, because this section contains much information including many references. To me, some of them are not "conclusion" of this study. Therefore, I strongly recommend that you change the title of this section to something like "Summary and conclusions".

25 We agree and changed the section name accordingly.